# Wind speed stilling and its recovery due to internal climate variability

Jan Wohland[1], Doris Folini[2,*], and Bryn Pickering[1,*]

[1]Climate Policy Group, Institute for Environmental Decisions, ETH Zürich, Zürich, Switzerland
[2]Institute for Atmospheric and Climate Science, ETH Zürich, Zürich, Switzerland.
[*]equal contribution

**Correspondence:** Jan Wohland (jwohland@ethz.ch)

**Abstract.** Near-surface winds affect many processes on planet Earth, ranging from fundamental biological mechanisms such as pollination to man-made infrastructure that is designed to resist or harness wind. The observed systematic wind speed decline up to around 2010 (stilling) and its subsequent recovery have therefore attracted much attention. While this sequence of downward and upwards trends and good connections to well established modes of climate variability suggest that stilling could be a manifestation of multidecadal climate variability, a systematic investigation is currently lacking. Here, we use the Max Planck Institute Grand Ensemble (MPI-GE) to decompose internal variability from forced changes in wind speeds. We report that wind speed changes resembling observed stilling and its recovery are well in line with internal climate variability, both under current and future climate conditions. Moreover, internal climate variability outweighs forced changes in wind speeds on 20 year timescales by one order of magnitude. Albeit smaller, forced changes become relevant in the long run as they represent alterations of mean states. In this regard, we reveal that land use change plays a pivotal role in explaining MPI-GE ensemble mean wind changes in the representative concentration pathways 2.6, 4.5, and 8.5. Our results demonstrate that multidecadal wind speed variability is of greater relevance than forced changes over the 21st century, in particular for wind related infrastructure like wind energy.

## 1 Introduction

According to station observations, near surface wind speeds declined between approximately 1980 and 2010, often referred to as stilling (Vautard et al., 2010). Land use changes were discussed as an important driver for the decline (e.g., Wever, 2012; McVicar et al., 2012; Zhang et al., 2019), implying that a reversal of land use changes would be needed to undo wind speed reductions. Over the last decade, however, wind speeds have increased without land use change reversal, potentially suggesting oscillatory behaviour of wind speeds rather than continuous decline (Zeng et al., 2019). In a dedicated modeling study that systematically sampled the realistic parameter space of, among others, roughness length and greenhouse gas concentrations, Bichet et al. (2012) found only small and sometimes insignificant effects of these forcings on wind speeds, supporting the notion of internal variability as the cause for stilling. Since multidecadal wind speed variability has direct implications for wind energy (Wohland et al., 2019b), an improved understanding of its causes would prove beneficial in locating and sizing wind power appropriately in addition to furthering conceptual understanding of climate dynamics. This paper therefore addresses whether stilling and its reversal are manifestations of internal climate variability or have been forced.

Adopting a longer term perspective allows us to contextualize the changes observed over the last four decades. Unfortunately, wind observations in the first half of the twentieth century are of little help in this regard owing to substantial changes in measurement techniques that gave rise to spurious trends (e.g., Cardone et al., 1990; Ward and Hoskins, 1996). Observations are also critically sensitive to anemometer height, as exemplified by different signs of the trends in 2m and 10m wind speeds in Australia (Troccoli et al., 2012). While satellites allow insightful investigation of wind speed variability over the ocean (e.g., Young and Ribal, 2019), they are only available over a short timespan of typically 30 years or less, making them effectively useless in the context of this study. Products that are partly based on satellite information, such as the modern reanalyses MERRA2 or ERA-interim, typically begin around 1980 and inter-reanalysis disagreement is documented in some parts of the world (Torralba et al., 2017). The new backward extension of ERA5 to 1950 might help to alleviate this shortcoming, however, it remains to be shown that the heavily evolving number and quality of observations has not induced spurious trends similar to the other ECMWF long-term reanalyses ERA20C and CERA20C. Centennial reanalyses provide long-term wind speed information that in theory should be consistent with the assimilated observations and underlying physics. In reality, however, strong discrepancies exist among current centennial reanalyses regarding long-term trends (Befort et al., 2016; Bloomfield et al., 2018) that are directly related to the assimilation of marine winds in the ECMWF products (Wohland et al., 2019a). Nevertheless, after trend removal, current centennial reanalyses consistently report multidecadal changes in German wind energy potentials that favour the interpretation of stilling as a phase in longer-term climate variability (Wohland et al., 2019b).

Global climate models allow us to complement observation-based and reanalysis-based assessments and can be used to evaluate internal variability versus forced changes under past and future climatic conditions. Despite their undisputed power in many applications, climate models are imperfect tools and uncertainties have to be properly accounted for. Following Hawkins and Sutton (2009), climate model uncertainties are often compartmentalized into model uncertainty (i.e., different implementations in different models), scenario uncertainty (i.e., uncertain forcings) and internal variability (i.e., inherent variability that can mask or amplify forced changes). Ensembles of different climate models, such as the ones contributing to the Climate

Model Intercomparison Projects (CMIP, Taylor et al., 2012; Eyring et al., 2016), can be used to quantify model uncertainty and derive results that are robust across the ensemble. CMIP5 models have been repeatedly used to investigate properties of past and future winds in the context of wind energy (e.g., Reyers et al., 2016). Scenario uncertainty can be overcome by investigating multiple plausible futures, rather than trying to accurately project the *real* future evolution of the climate system.

The role of internal climate variability can be quantified using large ensembles of the same climate model run with the same forcing that was initialized with different starting conditions (Maher et al., 2019). As a consequence of the different starting conditions, internal variability is generally out of phase in different ensemble members. When averaging over a sufficiently large ensemble, only those components that are synchronous in the ensemble (i.e., the forced signal) remain while internal variability cancels out. In this study, we will use the MPI Grand Ensemble (MPI-GE) to quantify the likelihood of stilling-like phases under past, present and future climate conditions and disentangle the effect of land use changes from those changes that are caused by elevated GHG concentrations. While land use change locally affects surface roughness and consequently wind speeds, altered GHG concentrations modify the global energy budget and impact the large-scale circulation. In this study, we address model uncertainty by comparing with the CMIP6 ensemble, scenario uncertainty by evaluating multiple scenarios and internal variability by analyzing the large ensemble MPI-GE.

## 2 Data and Methods

We mainly base our analysis on the Max Planck Institute Grand Ensemble (MPI-GE, Maher et al., 2019) and complement it with a large set of pre-industrial control simulations from CMIP6 (Eyring et al., 2016). MPI-GE provides a 2000y pre-industrial control simulation and 100 member ensembles for the historical period, three representative concentration pathways (rcp2.6, rcp4.5, rcp8.5), and a stylized scenario in which $CO_2$ concentrations increase by 1% per year while all other boundary conditions are kept unchanged. The 100 member ensembles are initiated from different years of the pre-industrial control simulation, sampling many different initial climate states. MPI-GE generally uses CMIP5 forcing (Taylor et al.), including land use data from Land Use Harmonization (LUH, Hurtt et al., 2011). Out of the total 500 ensemble members (5 experiments times 100 members), three members[1] were excluded from the analysis as a cautionary measure because they had a dozen duplicate time steps.

To meaningfully compare with observations and prior work, we investigate 10m wind speeds. So far, climate change impacts on wind energy were usually computed based on extrapolated near surface winds (e.g., Pryor et al., 2020; Schlott et al., 2018; Tobin et al., 2016; Karnauskas et al., 2018; Wohland et al., 2017) using logarithmic or power law relationships (Emeis, 2018). Since near-surface wind speeds have been the starting point in wind energy climate impact studies, an in-depth investigation of near-surface winds is pivotal to contextualize prior studies. In MPI-GE, wind speeds are computed in the model every 450 seconds from the wind components, minimizing the effect of canceling wind components during an averaging window. Model output is available as monthly means and we average to annual means as alterations of the seasonal cycle are not relevant here.

[1] rcp26_r055i2005p3, rcp45_r007i2005p3 and rcp45_r021i2005p3

Different scenarios have different forcings. In the pre-industrial control simulation, forcing is absent and time series exclusively represent internal variability. In contrast, forcings such as changing greenhouse gas concentrations and land use change exist in the historical and rcp experiments, and time series represent a superposition of internal variability and a response to the forcings. In the 1%CO2 experiment, forcing is exclusively due to evolving $CO_2$ concentration (i.e., land use is kept at its 1850 state) and time series represent a superposition of internal variability and a response to the $CO_2$ forcing. Given the relatively large ensemble, we consider the ensemble mean as a reasonable proxy for the forced response and calculate internal variability $s_i$ of each ensemble member by subtracting the ensemble mean $\langle s_i' \rangle$

$$s_{\mathrm{i}} = s_{\mathrm{i}}' - \langle s_i' \rangle, \tag{1}$$

where $s_{\mathrm{i}}'$ is wind speed in ensemble member $i$.

In this study, we calculate changes in wind speed $\Delta s$ averaged over a decade to mute interannual variability which has been studied elsewhere. For example, in the historical period, forced wind speed changes are computed as the difference between ensemble-mean wind speeds averaged over the last decade $s_{\mathrm{end}}$ (1990 to 2000) and first decade $s_{\mathrm{start}}$(1850 - 1860)

$$\Delta s = s_{\mathrm{end}} - s_{\mathrm{start}} \tag{2}$$

When evaluating future experiments, we report changes in 2090 to 2100 relative to 1850 to 1860, unless stated differently. To gain process insight, we decompose changes in ensemble mean wind speeds into a component due to the dynamical response to greenhouse gas forcing $\Delta_{\mathrm{dyn}} s$ and a residual $\Delta_{\mathrm{res}} s$ as

$$\Delta s = \Delta_{\mathrm{dyn}} s + \Delta_{\mathrm{res}} s, \tag{3}$$

in all locations that contain either primary or secondary vegetation.

As an estimate of the dynamical contribution $\Delta_{\mathrm{dyn}} s$, we compute wind speed changes in the idealized 1%CO2 simulation. We compare the decade with $CO_2$ concentrations equivalent to the end of the other experiments $s_{1\%\mathrm{CO2}}(\boldsymbol{x})(t_e)$ and the first decade of the simulation $s_{1\%\mathrm{CO2}}(\boldsymbol{x})(1850)$:

$$\Delta_{\mathrm{dyn}} s(\boldsymbol{x}) \approx s_{1\%\mathrm{CO2}}(\boldsymbol{x})(t_e) - s_{1\%\mathrm{CO2}}(\boldsymbol{x})(1850) = \Delta s_{1\%\mathrm{CO2}}(\boldsymbol{x})(t_e), \tag{4}$$

where $\boldsymbol{x}$ denotes location, and $t_e$ is given in Table 1. This approach allows us to separate $CO_2$ forced changes (computed from the stylized 1%CO2 simulation) from changes due to all forcings (based on the historical and rcp simulations). While this approach provides a reasonable proxy, it is not exact for a few reasons, including the effect of non-$CO_2$ species such as ozone and the stronger emissions in the stylized experiment that leaves the climate system less time to respond to the forcing. We compare the residual changes $\Delta_{\mathrm{res}} s$ to the changes in primary and secondary land in the LUH1 dataset to quantify the effect of land use change.

## 2.1 Trends

We compute trends over 20y time periods which is a reasonable timescale for stilling and its reversal given that stilling in the observational datasets spans 25 to 30 years while its reversal currently lasts less than 15 years (Zeng et al., 2019). We deter-

**Table 1.** $CO_2$ concentration at the end of each experiment, taken from the IPCC's 5th Assessment Report (Table AII.4.1 on p. 1422, Stocker, 2014) and rounded. End decade denotes the last decade that is completely contained in an experiment. 1%CO2 equivalent year corresponds to the year in which the same $CO_2$ is achieved following a 1% per year growth trajectory that starts in 1850.

| experiment | end decade | final $CO_2$ concentration | 1%CO2 equivalent year $t_e$ |
|---|---|---|---|
| historical | 1990 - 2000 | 380 ppm | 1881 |
| rcp85 | 2090 - 2100 | 935 ppm | 1971 |
| rcp45 | 2090 - 2100 | 540 ppm | 1916 |
| rcp26 | 2090 - 2100 | 420 ppm | 1891 |

mine statistical significance based on a Wald test with t-distribution of the test statistic as implemented in scipy.stats.lingress, identical to the approach taken in Wohland et al. (2020).

The historical (future) period contains 155 (95) years per ensemble member and we perform trend analysis for all complete consecutive 20y periods. As a consequence, adjacent periods are not independent but share 95% (19y/20y) of data. We ensured, however, that clustering of trends does not relevantly impact the results by visual inspection of the timing of significant trend periods. Our approach yields 135 (75) trend estimates per ensemble member, totaling 13500 (7500) for the whole ensemble, allowing to robustly investigate the role of multidecadal internal variability under historical and future climate conditions.

## 3 Results

We report results in two parts. In Sec. 3.1, we investigate the relative importance of land use change and altered $CO_2$ concentrations in explaining forced wind speed change and report that land use change plays a pivotal role in explaining periods of exceptionally high (historical 1950 - 2000) and low (rcp45 2050 - 2100) wind speeds. Later, in Sec. 3.2, we turn to multidecadal variability in individual ensemble members and find that stilling and reversal-like periods occur frequently and as a consequence of internal climate variability in all scenarios.

### 3.1 Decomposition of forced wind speeds into a dynamical and a residual contribution

Figure 1 depicts ensemble mean changes in wind speeds for the historical period and three future scenarios. By taking the mean over the 100 ensemble members, internal variability is substantially reduced, effectively leaving only the forced component behind. As detailed in the Methods section, we decompose the full change (1st column; a,e,i,m), into a dynamical change due to increased $CO_2$ concentrations (2nd column; b,f,j,n) and the residual (3rd column; c,g,k,o).

Broadly speaking, the dynamical changes dominate offshore while the residual changes are strongest over land. Dynamical changes are relatively weak in the historical and rcp26 experiments and, as expected, become stronger with increasing $CO_2$ forcing in the rcp45 and rcp85. One very prominent feature is an intensification of wind speeds over the Southern Ocean while over land dynamical wind changes are weakly negative over the historical period and become more negative with increasing $CO_2$ concentrations. Onshore wind speed changes are predominantly positive and distinct hotspots exist, for example, in the central United States, along the coasts of Southeastern South America and Southeastern Africa, and in Eastern Europe. These regions not only experienced wind speed increases but also decreases in primary and secondary land. In fact, there is a good connection between land use change and residual wind speed changes in many locations.

In-depth analysis strengthens the conclusions that can already be drawn from visual inspection. Averaging over all onshore locations (excluding the major ice sheets in Antarctica and Greenland), confirms that average dynamical wind speed changes are negative (see Fig. 2). Their amplitude is small (mean around -0.1m/s) in the historical period and a change smaller than -0.25 m/s almost never occurs (Fig. 2a). In the rcp85, in contrast, the area mean change is around -0.25 m/s and even values smaller than -0.5 occur frequently (Fig. 2d). Changes in the residual wind speeds, however, are predominantly positive and of similar amplitude as the dynamical changes. Land use and dynamical changes thus partly offset each other. Residual changes are comparable in the historical experiment and rcp26, while they are twice as strong in rcp85 and disappear on average in rcp45. The different evolution in rcp45 is linked to an increase in primary and secondary land that is discussed later (see Fig. 4). While the distribution of dynamical changes is narrow in the historical periods, it substantially widens with $CO_2$ emissions.

**A strong link between land use change and residual wind speed change**

Moreover, there is a very clear link between land use change and change in residual wind speeds at those locations where the residual wind speed change is positive (Fig. 3). The conditional probability of observing a negative change in primary and secondary land given a non-negative change in residual wind speed $p(luc < 0|\Delta_{\mathrm{res}} s' \geq 0)$ is always higher than 95% in all

experiments. With increasing changes in residual wind speeds, it quickly becomes indistinguishable from 100%. In addition, correlations between land use change and residual wind speed change are considerable (Pearson correlation coefficient around -.6 and Spearman rank correlation between -.48 and -.64). A simple binary classifier that predicts the sign of land use change as the negative sign of residual wind speed change is correct in 71% (historical and rcp26), 66% (rcp45) and 82% rcp85 of all locations.

While these numbers clearly document a link between residual wind speed changes and changes in primary and secondary land, a one-to-one relationship does not exist. Such a relationship, however, is also not expected for two main reasons. First, the effects of land use changes are not restricted to the immediate vicinity and wind speeds in one grid box may well be influenced by land use changes in adjacent grid boxes. Second, the correspondence between the idealized 1%$CO_2$ experiment and the other experiments is only an imperfect proxy for the dynamical change due to, among others, a significantly higher rate of emissions that leaves the climate system less time to respond to the forcing. Another important difference are non-$CO_2$ emissions like ozone and aerosols that are ignored in the idealized 1%$CO_2$ experiment. Nevertheless, we report a very good link between positive residual wind speed changes and land use change overall.

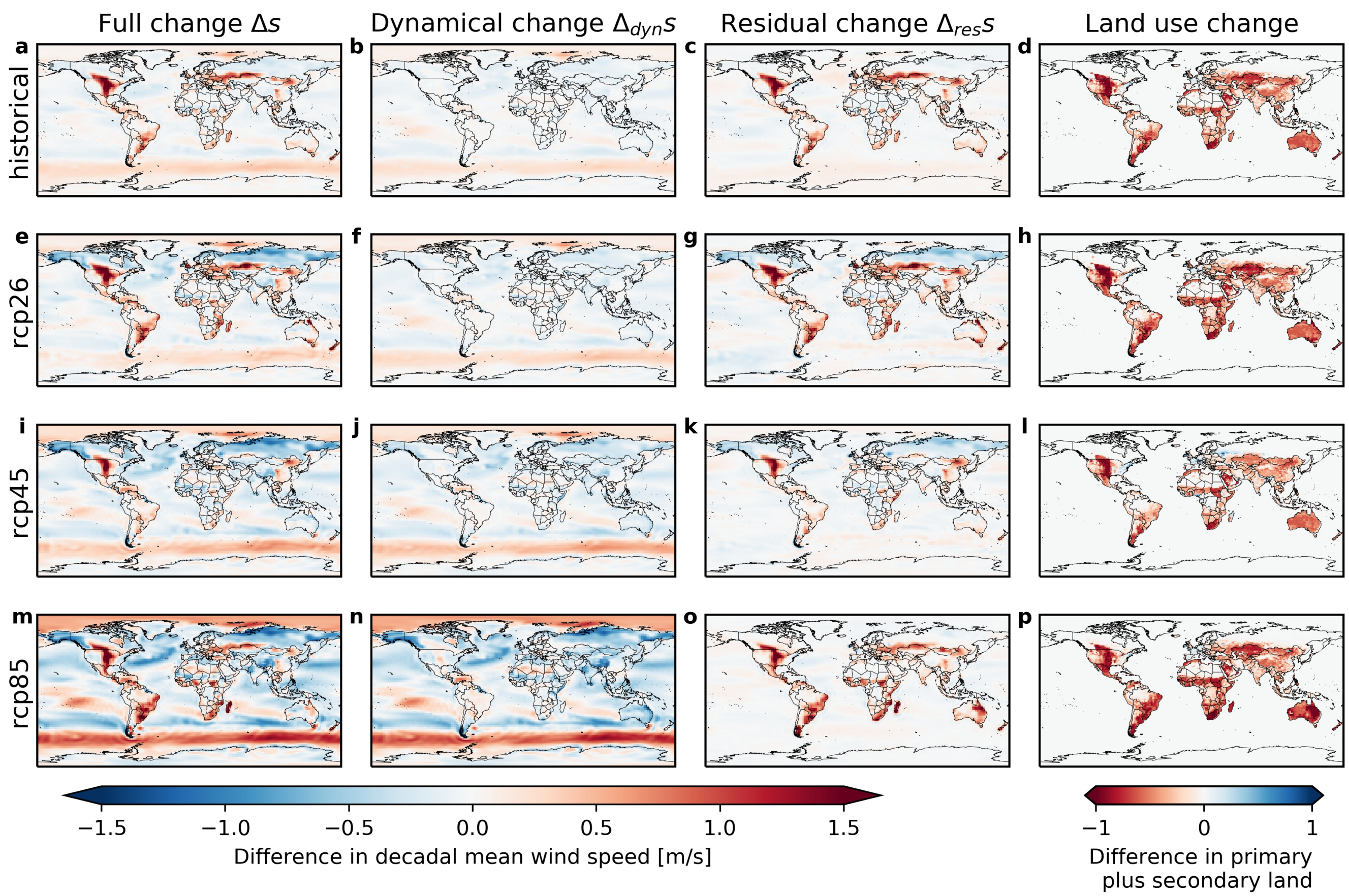

**Figure 1. Near-surface wind speed changes in the MPI-GE ensemble mean and change in primary plus secondary land in LUH1**. Changes are expressed relative to the beginning of the historical period (1850-1860). Each row represents one experiment (historical, rcp2.6, rcp4.5, rcp8.5) and values are shown for the last complete decade in each experiment (i.e., 1990-2000 in historical and 2090-2100 for the rcps). The first column shows the full change (i.e., the change in wind speeds in the respective experiment), second column shows the dynamical change due to $CO_2$ forcing only (calculated from the 1%CO2 experiments during decades including the years given in Table 1), third column shows the difference between the 1st and 2nd column which we refer to as residual. The last column denotes land use change.

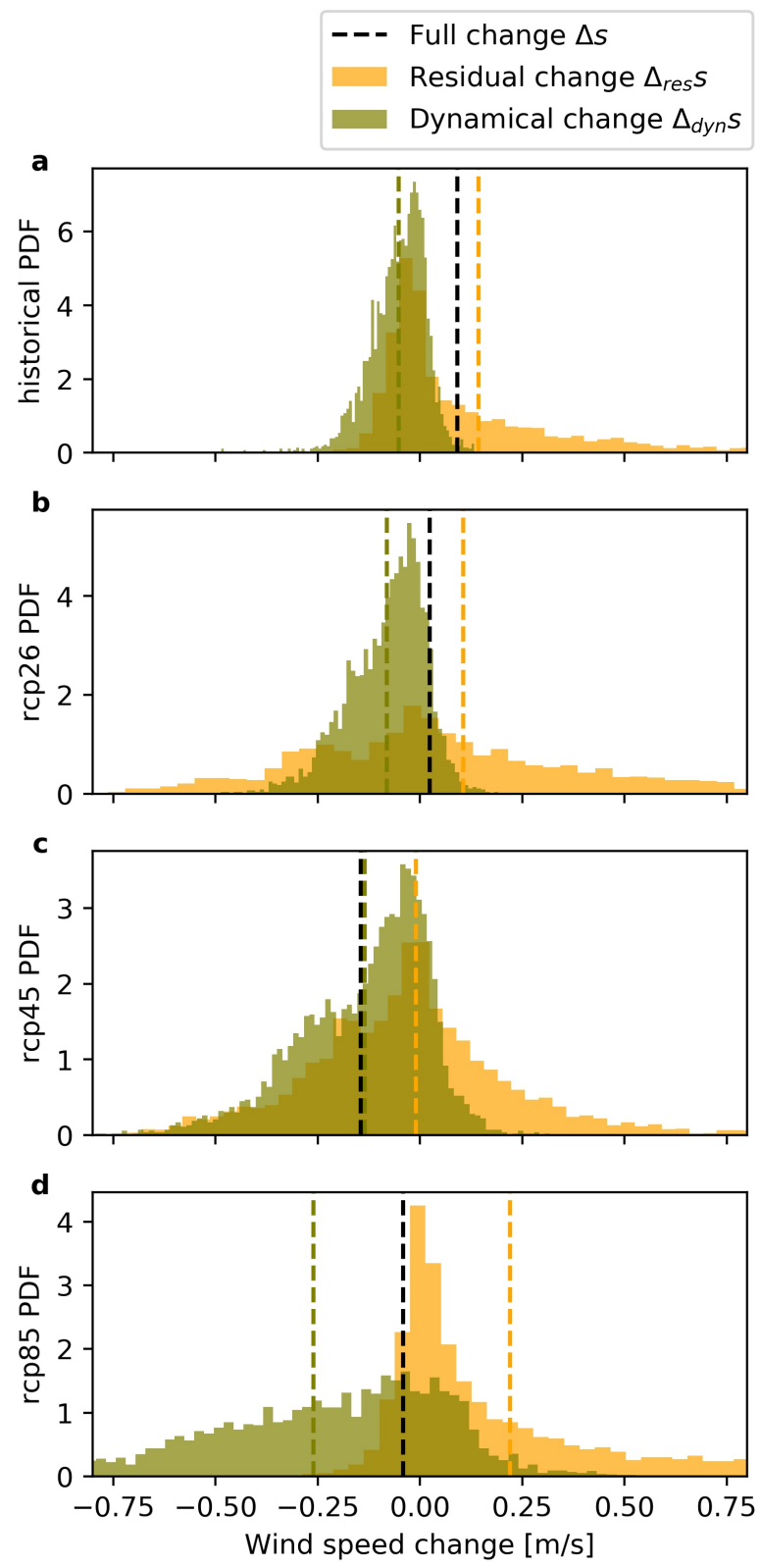

**Figure 2. Probability density function of wind speed changes over land (excluding Antarctica and Greenland).** Dashed lines mark the mean change. Values correspond to columns 2 and 3 in Fig. 1.

.

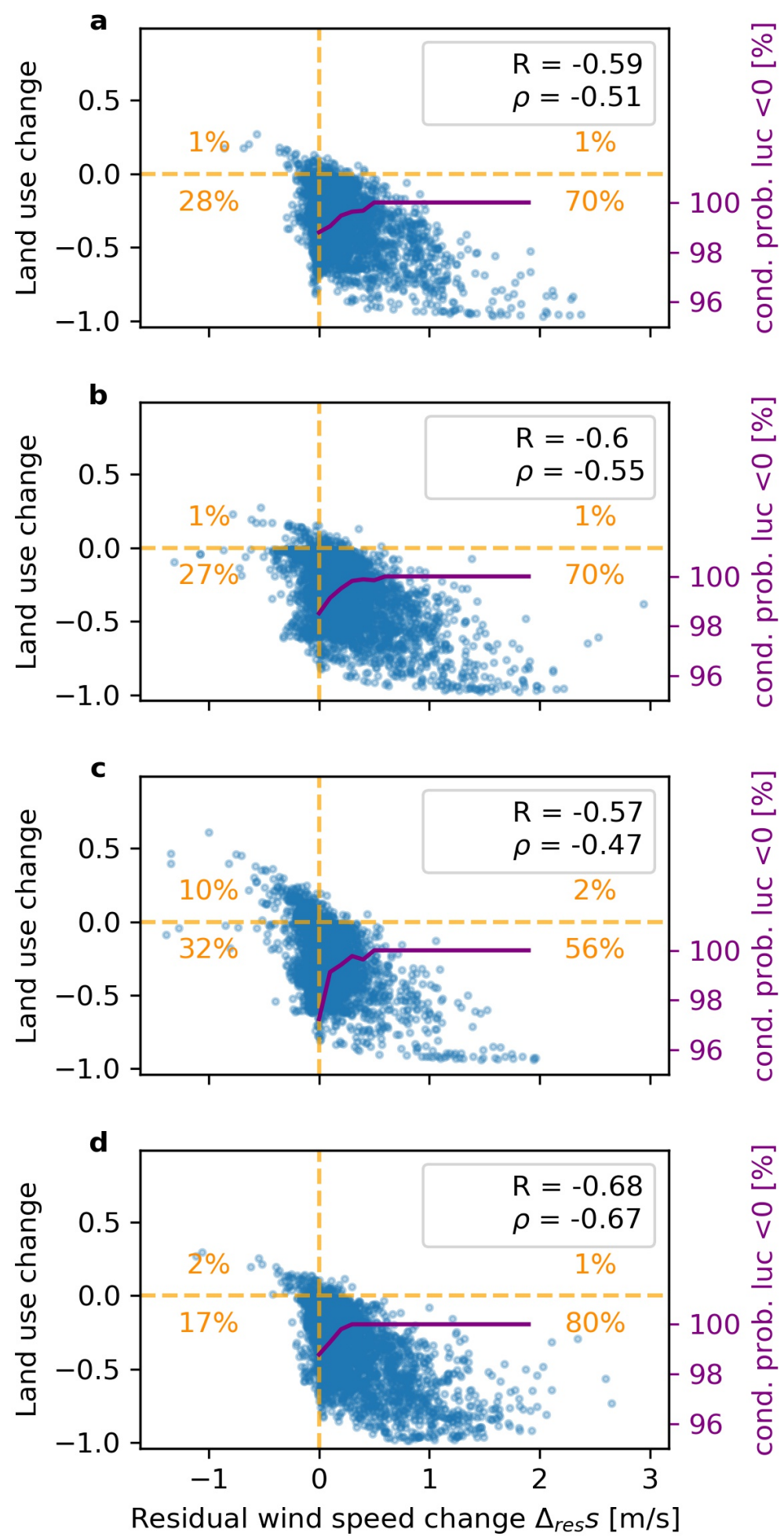

**Figure 3. Scatter plots between changes in residual wind speeds and land use change over all locations with non-zero land use change (absolute value greater than 0.01)**. Each subplot denotes one scenario: historical (a), rcp2.6 (b), rcp4.5 (c), rcp8.5 (d). The secondary purple y axis shows the conditional probability of negative land use change given a wind speed change equal or greater the corresponding value in the plot ($p(luc < 0|\Delta_{\text{res}}s > x)$). Pearson correlation coefficient $R$ and Spearman rank correlation $\rho$ are given in the legend.

### 3.1.1 Land use change often masks climate change impacts on future onshore winds

While the mean dynamical wind speed response is monotonous in the $CO_2$ forcing (small for rcp26, large for rcp85), no such link exists for the residual wind speeds (see Fig. 2). Instead, rcp45 features almost no change in residual wind speeds while rcp26 features a weak increase and rcp85 has a strong increase. This seeming discrepancy can be explained in terms of the land use scenarios: Comparing the late 21st with the late 20th century, rcp45 features dominantly increases in primary plus secondary land while the other scenarios dominantly show decreases (see Fig. 4). These increases in rcp45 lead to wind speed reductions that on average undo the changes during the historical period and thus yield a net zero change of residual wind speeds between 2090-2100 and 1850-1860 (Fig. 2c).

Comparing the dynamical and residual means from Fig. 2, we find that combining land use forcing from rcp45 with greenhouse gas concentrations from rcp85 would lead to forced wind speed reductions approximately twice as large as in rcp45. Such considerations are worthwhile because the generation of the representative concentration pathways is not an exact science but rather represents storylines that are considered plausible (van Vuuren et al., 2011). The above mentioned scenario merger could, for example, occur if humanity aims to reduce emissions and values reforestation highly (as foreseen in rcp45) while fossil fuels are still dominantly used.

In real world applications, the combined effect of all forcings matters. Therefore, we plot the onshore-mean ensemble-mean wind speeds in Fig. 5. We first want to note that the values match well with observations, suggesting small biases. For instance, Wu et al. (2018) report wind speeds in the range of 3.31 m/s to 3.5 m/s for the global mean excluding Australia and a lower value of around 2.1 m/s in Australia, both during 1981-2010. Calculating the area averaged wind speed (5% land in Australia), yields 3.25 to 3.43 m/s, which includes MPI-GE ensemble mean estimate of around 3.36 m/s.

With respect to the temporal evolution, two distinct phases can be identified. During the historical period, there is a clear upward tendency in line with the reduction of primary and secondary land, as discussed earlier. Approximately in 1950, global mean onshore wind speeds exceed the highest value that ever occurred in the 2000y pre-industrial control run. In other words, the ensemble mean wind speeds in 1950 had less than 0.05% likelihood of occurrence in one ensemble member under pre-industrial climatic conditions. The fact that such high wind speeds occur in a 100-member ensemble mean, further reduce the likelihood (by a factor of 1/100 if the ensemble members are considered fully independent), making it virtually impossible to obtain such high values under pre-industrial conditions. Nevertheless, values are even higher on average until around 2010.

In around 2000, the tendency inverts and wind speeds begin to decline. The decline is strongest in rcp45, dropping below the calmest pi-control year from around 2050 onwards. This strong reduction is a combination of residual wind speed changes that were positive in 1990-2000 and are effectively zero in 2090-2100 (both compared to 1850-1860), and a dynamical contribution that reduces wind speeds on average. Rcp26 and rcp85 are within the range seen under pre-industrial conditions and are much closer to 1850-1860 average wind conditions. In rcp26 positive wind speed changes due to reduced primary and secondary land slighlty outweigh dynamical wind speed reductions while the opposite is the case in rcp85. Overall, we have shown that global mean onshore wind speeds have left the ranges experienced under pre-industrial climate conditions in the second half

of the 20th century and will leave them again in 2050 in the rcp45 scenario. The future ensemble-mean evolution is to a large extent governed by land use change which compensates to varying degrees the $CO_2$ induced changes.

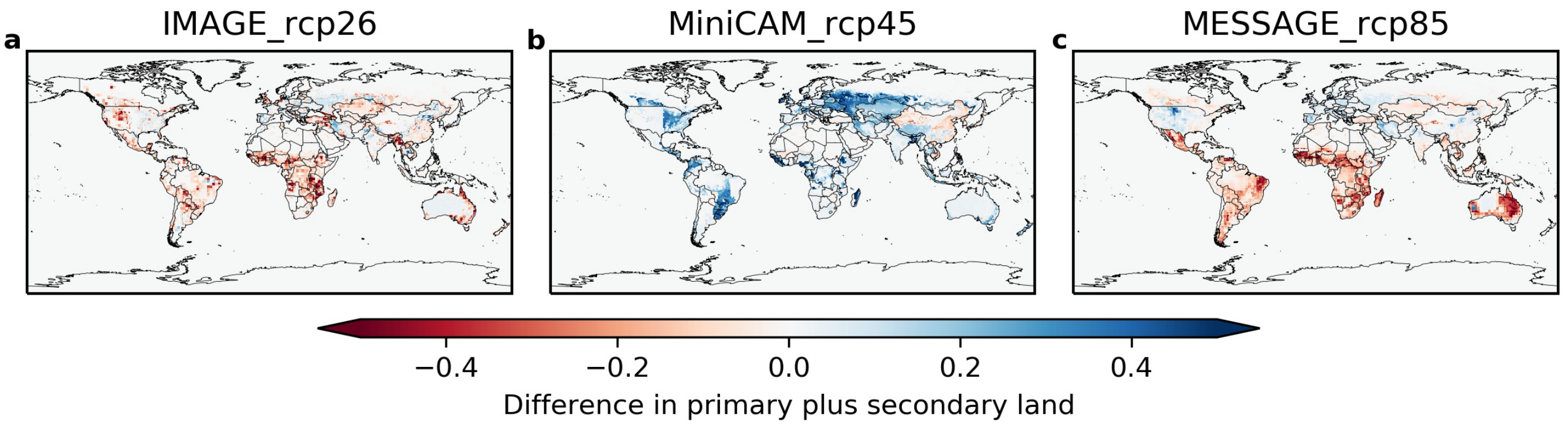

**Figure 4. Change in primary plus secondary land in the rcp scenarios.** Maps show difference between 2090-2100 and 1990-2000 according to LUH. Values are given as fraction of the land that changes its land use classification.

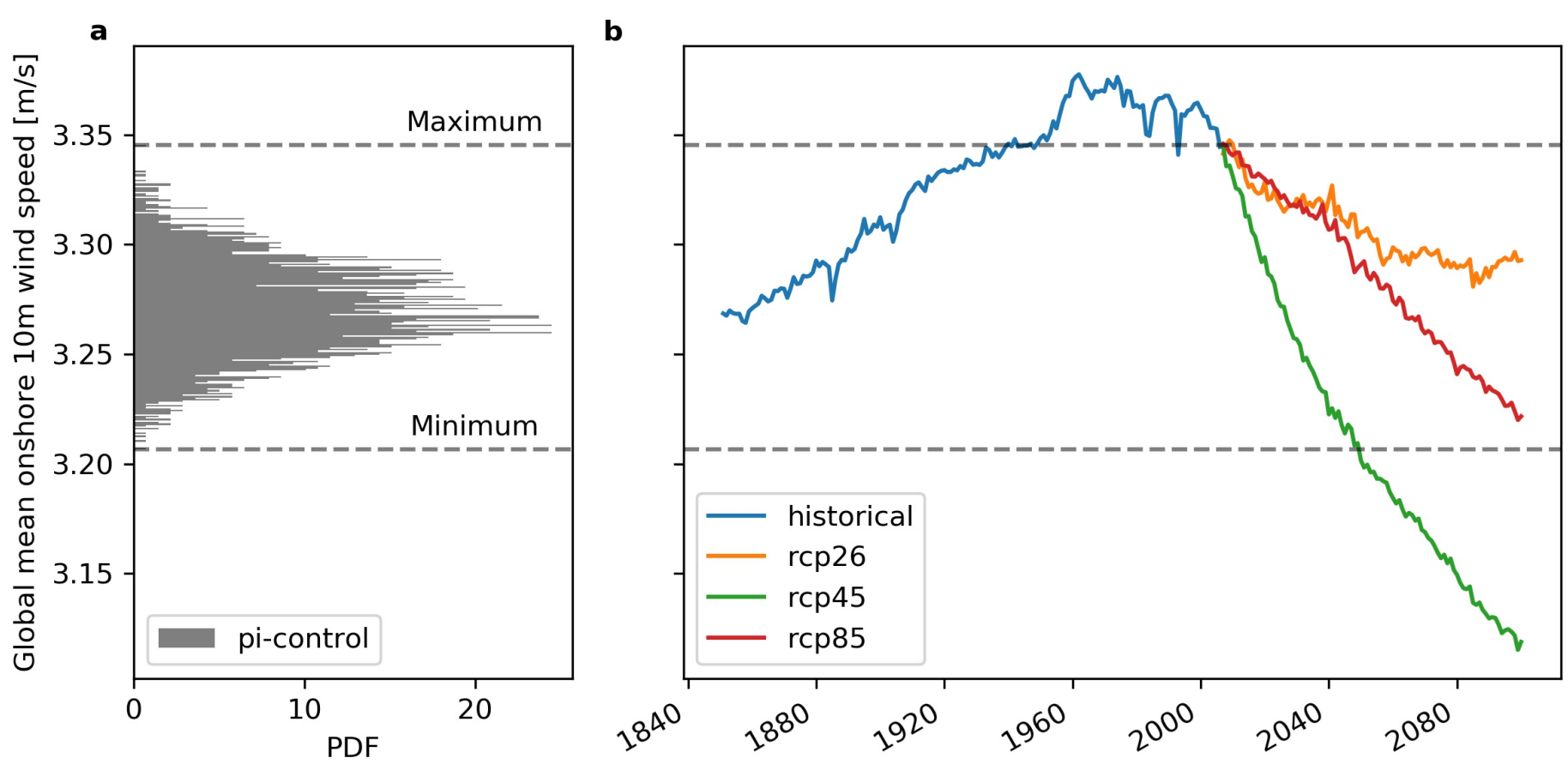

**Figure 5. Global mean onshore wind speeds in MPI-GE.** a) Histogram of pi-contril annual mean wind speeds (flipped). b) ensemble-mean wind speed evolution during historical (blue) and rcp (red, orange, green) scenarios.

### 3.2 Internal wind variability consistent with observed stilling and its recovery in Europe

So far, we have focused on forced changes that can readily be detected in a large ensemble by investigating alterations of the ensemble mean. We will now add an analysis of internal climate variability by evaluating fluctuations around the ensemble mean. Here, we report values for Europe (defined as a rectangular box covering 37.5°N to 60°N and 10°W to 25°E, see inset in Fig. 6) and also compare to values obtained by interpolation to the European HadISD station sites. This second step is important to understand the relative magnitude of forced changes as compared to variability and it is needed when comparing to observations because reality only provides a single realization in which we can take measurements.

Fig. 6 shows wind speeds averaged over Europe during the pre-industrial control simulation. We added red and green markers to denote periods in which statistically significant 20y trends begin. Such trend onsets occur repeatedly during the 2000y simulation despite the absence of a long-term trend or any forcing.

We present histograms of statistically significant trends in Fig. 7. The distribution is bimodal with a peak at approximately ±0.1m/s/decade which fits very well to the trends reported in (Vautard et al., 2010; Zeng et al., 2019), suggesting that both stilling and its reversal are of a magnitude that can be explained by internal variability alone. Moreover, stilling and its reversal are frequent events. Almost half of the time (45%), wind speeds are either in a significant downward or upward trend.

These results are robust and remain valid under different spatial sampling, different trend lengths and using a large CMIP6 ensemble. For instance, when evaluating wind speeds interpolated to all observational sites in Europe (see Fig. A2), trends remain largely unchanged; we present trends averaged over the European box in the remainder of this section for simplicity. As expected, trends computed over a shorter period occur more often (57% for 15y trends) and have a larger magnitude while longer trends have weaker magnitudes (see Fig. A3). While the specifics vary with trend length, significant trends of similar magnitudes to the observed ones emerge independent of trend length. To also account for model uncertainty, we repeated the assessment with the full CMIP6 model ensemble that was available in early February 2021. Out of the 55 available models, two were excluded because they provided no data (EC-Earth3-CC) or showed a suspiciously constant drift (AWI-ESM-1-1-LR), and the ensemble mean was calculated using the remaining 53 models (1 realization per model). For each model, we computed a trend histogram and provide the (equally weighted) mean over all ensemble PDFs in 7b. Even though the ensemble mean PDF peaks at slightly lower values, it is consistent with the interpretation that stilling and its reversal can be manifestations of internal climate variability. Analysing trend PDFs of individual models, we find that a large subset agrees remarkably well with the amplitudes reported by MPI-GE, while some models feature lower trend magnitudes (see Fig. A4).

#### 3.2.1 Internal wind variability dominates forced trends 10-fold under historical and future climate conditions

While we have shown that stilling-like trends occur under pre-industrial control conditions, it remains possible that observed stilling was a combination of a forced change and internal variability. In fact, Fig. 5 suggests that global wind speeds peaked in 1980 or so and declined afterwards, seemingly supporting this interpretation. Moreover, it is conceivable that forced changes not only dominate over internal variability but also alter internal variability.

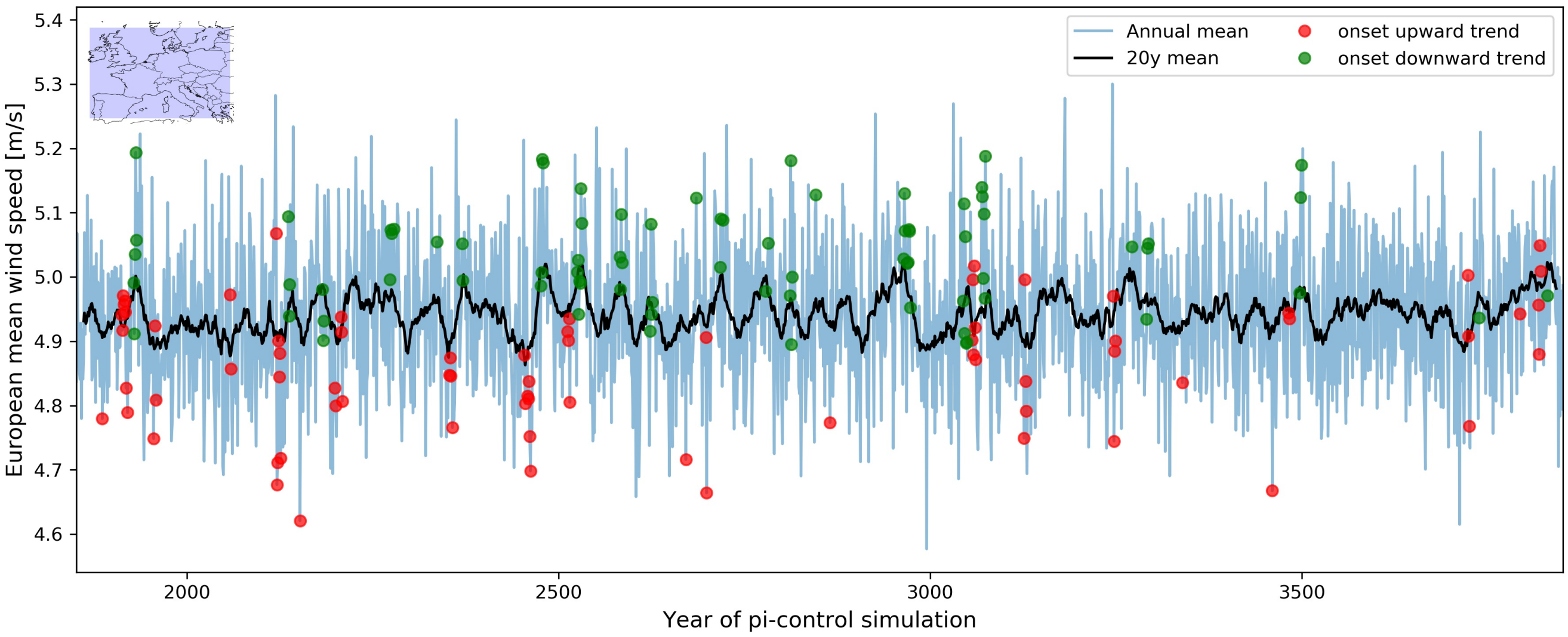

**Figure 6. Wind speed timeseries during pi-control averaged over Europe.** Blue (black) lines denotes annual (20y) means and red (green) dots mark onset years of statistically significant upward (downward) trends over 20y periods.

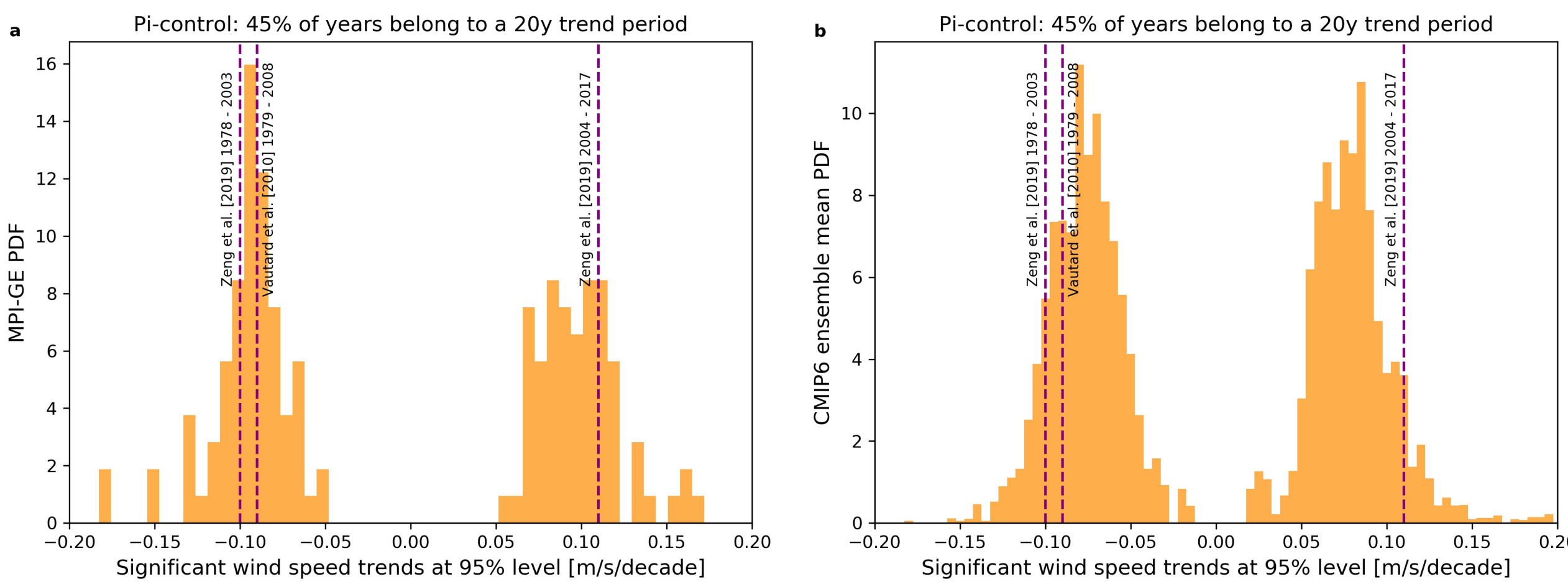

**Figure 7. 20-year trends in European annual mean wind speed in MPI-GE (left) and the CMIP6 multi-model mean (right)**.Trends are only shown if they are different from zero at a 95% significance level. The CMIP6 histogram is the mean over the trend histograms of the different CMIP6 ensemble members. Two models were excluded from the ensemble mean due to strong model drift (AWI ESM-1-1 LR) and missing data (EC-Earth3-CC), however there are remaining ensemble members from the two model families.

Evaluating the MPI-GE ensemble, however, we find that 20y trends of the forced components are too weak to explain stilling roughly by a factor of ten (see Fig. 8). For example, in the historical period, the trends in the forced wind speed changes is

at the order of 0.01 m/s/decade while the observed trends are one order of magnitude larger. This discrepancy is in line with the results of Bichet et al. (2012) who report that observed trends are 5 to 15 times larger than trends obtained from varying greenhouse gas concentrations, land use change, aerosols and other parameters. Similar trend values (of different signs) are also found for all future scenarios studied here. It should be noted, however, that some of the trends in the forced timeseries are predominantly (historical, rcp26) or exclusively (rcp45, rcp85) of the same sign. They thus become very important over longer periods covering many decades or centuries.

Moreover, we report that the characteristics of trends that occur due to internal variability is not strongly impacted by the chosen experiment. This is true for the likelihood of statistically significant 20y trend periods which is always close to 50% and the shape of the trend PDFs which always peak around 0.1m/s/decade and -0.1m/s/decade.

To summarize, 20y wind speed trends occur frequently under current and future climate conditions as a consequence of internal climate variability. Their amplitude is similar to that found in observations, suggesting that stilling and its recovery are manifestations of climate variability. Forced trends in wind speeds are too weak (by approximately one order of magnitude) to explain stilling and its recovery, according to MPI-GE and backed up with a large CMIP6 ensemble.

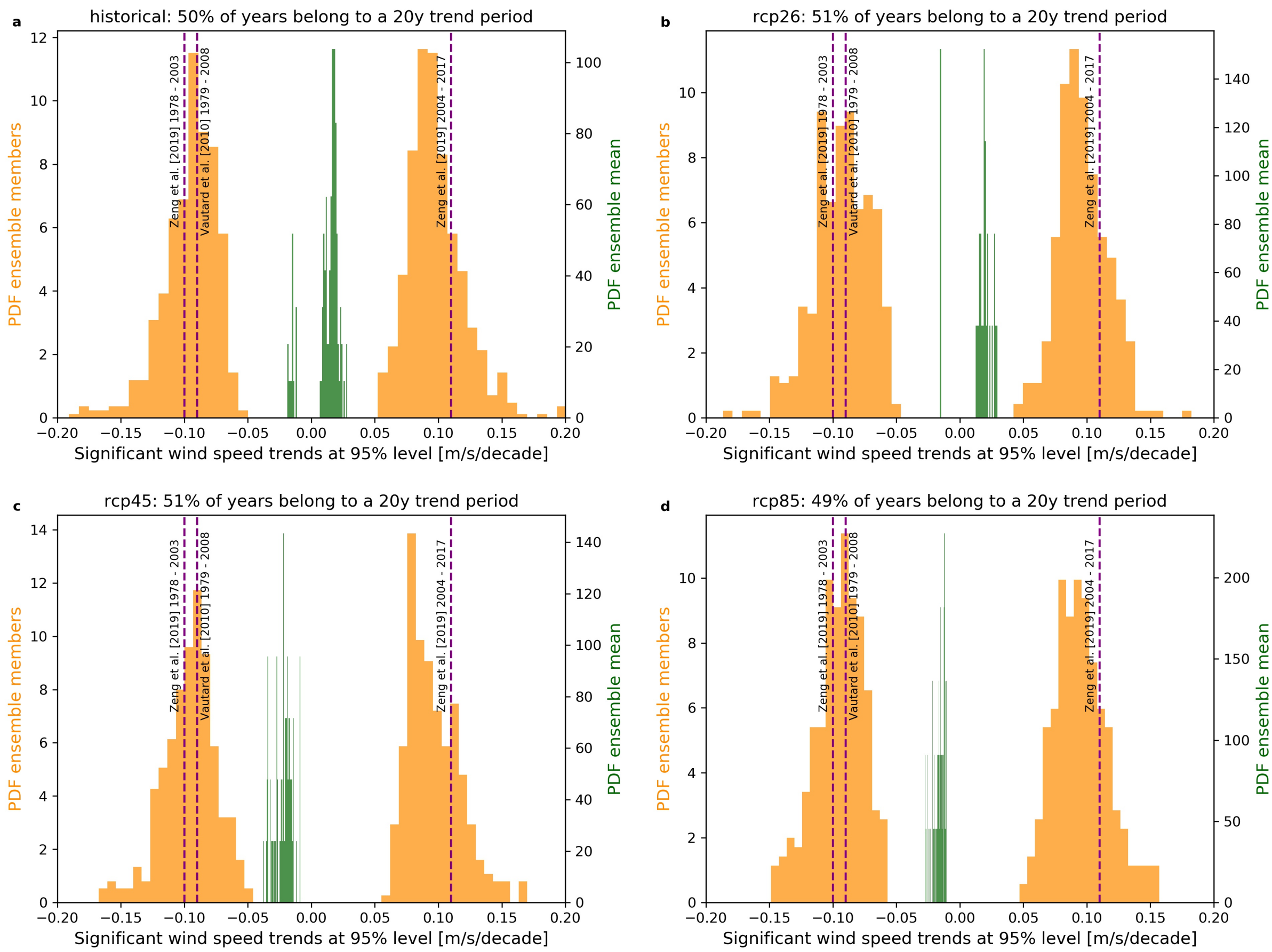

**Figure 8. 20-year trends in European annual mean wind speed in MPI-GE under historic and future climate conditions**. Trends are computed for each ensemble member after subtraction of ensemble mean (yellow) and for the ensemble mean (green). Different subplots show different experiments. Trends are only shown if they are different from zero at a 95% significance level.

## 4 Discussion and conclusion

Drawing from the 100-member MPI Grand ensemble and its LUH1 land use forcing, we decomposed surface wind speed changes by cause. We have shown that land-use related changes play a dominant role over the historical period, leading to global mean onshore wind speeds in the late 20th century that are unprecedented in the unforced 2000y simulation. In experiments of future conditions, average wind speed reductions caused by higher GHG concentrations almost cancel wind speed increases due to land use change in rcp26 and rcp85, yielding wind speeds similar to those under pre-industrial conditions. Particularly strong increases in primary and secondary land, however, lead to record low wind speeds in rcp45 which consistently fall below the lowest values seen in pi-control in every year starting around 2050. Even though land use is a significant contributor to forced (i.e., ensemble mean) changes, it plays a minor role in understanding the stilling phenomenon. While internal climate variability frequently induces 20y trends of the same magnitude as observed stilling and its recovery, forced trends on such time scales are substantially smaller in the ensemble mean. Moreover, we find that stilling-like periods will continue to occur under future climate conditions (rcp26, rcp45, rcp85) in approximately 50% of all years independent of the applied greenhouse gas and land use forcing.

To the best of our knowledge, this study is the first to use a large climate model ensemble to understand the wind speed effects of land use change and wind speed stilling. Our results complement, extend and partly contradict the existing literature that is based on other lines of evidence.

In line with our results, Pryor et al. (2020) have argued in a recent review that natural wind variability dominates over forced changes due to anthropenic climate change. They further argue that the attribution of wind speed changes based on comparing a relatively short time period is substantially complicated by low-frequency climate variability and report that it is currently unclear whether future climate change will lead to further stilling or increased windiness. As demonstrated in this paper, a clearer picture can be obtained by using a large ensemble to separate the forced component from low-frequency climate variability; the forced component can be further decomposed to distinguish between the effects of land use change versus changes in atmospheric circulation.

Moreover, Gonzalez et al. (2019) decompose changes in the CMIP5 ensemble into a large-scale and a local component using the rcp85 scenario. They find that near-surface wind speed changes "are more negative than would be expected from the large-scale circulation alone" which makes perfect sense given the wind speed reductions due to land use change (e.g., Fig. 2d). Moreover, Gonzalez et al. also report remarkably good ensemble agreement for this reduction which again aligns well with our results because the land use forcing is synchronous across all models, thus yielding high inter-model agreement.

In contrast to our results, Zhang et al. (2019) conclude that stilling is dominantly caused by surface friction and report a surface friction contribution of 125% in Europe (which compensates a negative contribution from turbulent frictions and is hence larger than 100%). These results seemingly contradict our findings, yet we believe that the contradiction can be resolved. The authors use the difference between observed station wind speed and modeled wind speeds based on measured pressure gradients as a proxy for surface friction and only use stations where both wind estimates co-vary well, implying that the model might be subject to a sampling error. More importantly, Zhang et al. (2019) use regression analysis instead of explicit modeling,

which can create artifacts. In fact, we also report that ensemble mean wind speeds over land decline globally (Fig. 5) and in Europe (Fig. A1) from approx. 1970 onward. In the single realization that we consider the real world (i.e., observations), stilling has occurred in the same period and regression analysis would yield high agreement. The problem, however, is that despite the good agreement in terms of the trends, the amplitudes of the trends do not match. Such a mismatch of amplitudes remains unnoticed in regression analysis because regression is invariant to multiplication with a scalar.

In the wider context of climate change impacts on wind energy, our results using the 1%$CO_2$ experiment support early conceptual arguments that global warming would reduce wind speeds through a reduction of meridional pressure gradients following increased warming at the poles relative to mid- and low latitudes (e.g., Klink, 2007). In non-idealized systems, however, current best knowledge suggests that there are so many interacting and competing processes that it is "unknown whether anthropogenic warming will result in stilling (decreases in wind speed) or increased windiness" (Pryor et al., 2020). In addition to the competing effects of land use change and greenhouse gas emissions examined here, Bichet et al. (2012) find that aerosol emissions play a significant role in some places.

We believe that our results have three implications of greater relevance. First, many studies have sought to understand climate change impacts on wind and wind power generation based on CMIP5 simulations and more work is underway using the next generation CMIP6. In doing so, it was often implicitly assumed that altered concentrations of greenhouse gases are the main driver in future scenarios, in particular in high emission scenarios such as rcp85. However, by showing that land use change plays at least a similarly large role in explaining forced changes, our results challenge this assumption. Given that different land use scenarios can share the same level of greenhouse gas emissions (and vice versa), it follows that changes in wind speeds and wind power generation can not be directly linked to a certain level of greenhouse gas emissions. Instead, many combinations are theoretically possible and some of them can lead to even greater wind speed changes. Second, wind speeds trends over two decades are manifestations of internal climate variability and will remain largely unchanged under future conditions. This finding is consistent with an emerging body of literature that links multidecadal changes in wind and wind power during the historical period to patterns of mulitdecadal climate variability such as the North Atlantic Oscillation (Wohland et al., 2020; Zeng et al., 2019).

The energy sector will benefit from this improved understanding of the long-term dynamics of wind speeds. Planning of future renewable energy systems must account for forced long-term trends in wind speeds, as well as multidecadal wind power fluctuation from internal climate variability. Our clear identification of the significantly greater strength of multidecadal fluctuations has particular implications on wind power, since the timescale of these fluctuations is of the same order as the timescale of wind power projects. To account for both dynamics, a greater range of model years and future climate change scenarios should be incorporated into energy system analysis; it is not sufficient to consider only recent decades in planning for the future. The result of such an approach will ensure robustness to known variability, and can even capitalise on the existence of large ensembles to further reduce uncertainty, as we have done here.

## 5 Code and data availability

The underlying data and code are publicly available. MPI-GE surface winds (variable nam "sfcWind") can be downloaded from https://esgf-data.dkrz.de/projects/mpi-ge/. LUH1 land surface data can be retrieved from https://luh.umd.edu/data.shtml#LUH1_Data. CMIP6 data was taken from the internal ETH IAC data pool, yet is also available from the ESGF via https://esgf-data.dkrz.de/search/cmip6-dkrz/. Code is written in Python and is maintained on github. The code will be available upon publication at github.com/XYZ.

*Author contributions.* JW performed the simulations, analyzed the data, produced all figures and wrote most of the manuscript. DF helped to develop the methodology and contributed to the discussion of the results. BP reviewed the code and helped to analyze the data. DF and BP contributed equally and are listed in alphabetical order of their surnames. All authors contributed ideas, gave feedback and substantially improved the manuscript.

*Competing interests.* The authors declare that no competing interests exist.

*Acknowledgements.* JW wants to thank Maria Rugenstein, Edouard Davin and David J. Brayshaw for interesting exchange. The authors owe Urs Beyerle gratitude for facilitating access to the CMIP6 data. We acknowledge the World Climate Research Programme, which, through its Working Group on Coupled Modelling, coordinated and promoted CMIP6. We thank the climate modeling groups for producing and making available their model output, the Earth System Grid Federation (ESGF) for archiving the data and providing access, and the multiple funding agencies who support CMIP6 and ESGF. We want to thank the Max Planck Institute for Meteorology for making MPI-GE available. JW is funded through an ETH Postdoctoral Fellowship and acknowledges support from the ETH foundation and the Uniscientia foundation.

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

**Appendix A: Additional figures**

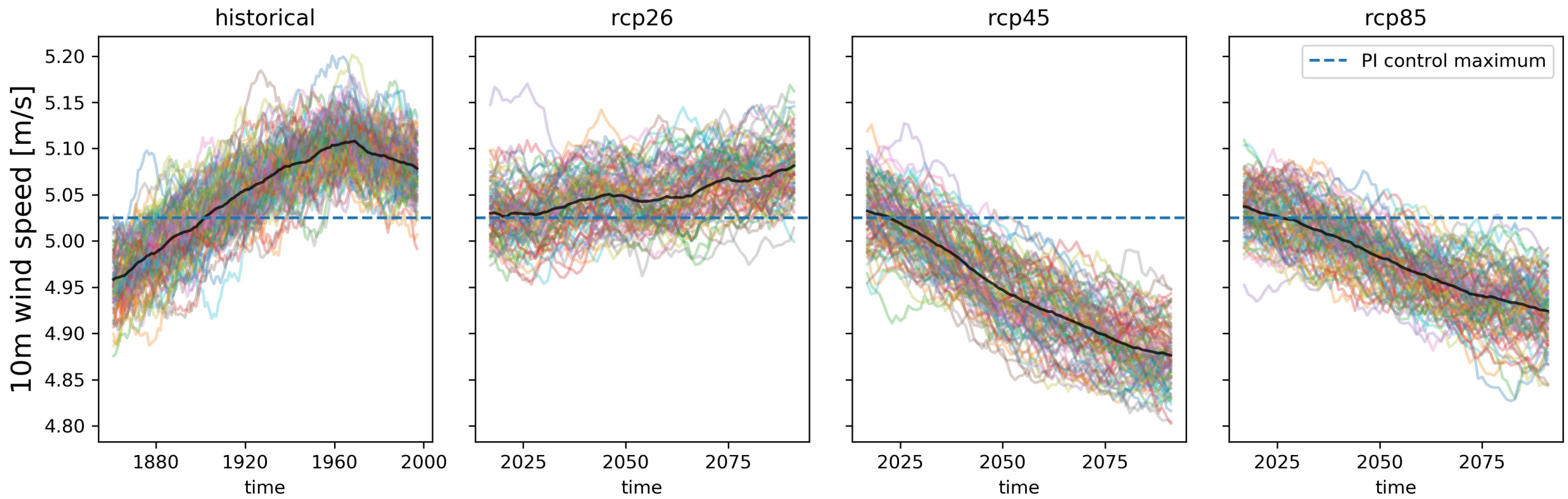

**Figure A1. Wind speed timeseries averaged over Europe for pi-control, historical and future experiments.** The dashed horizontal line denotes the maximum 20y mean value seen in the 2000y pi-control simulation. Ensemble memebers are given in colors and the ensemble mean is plotted as black solid line.

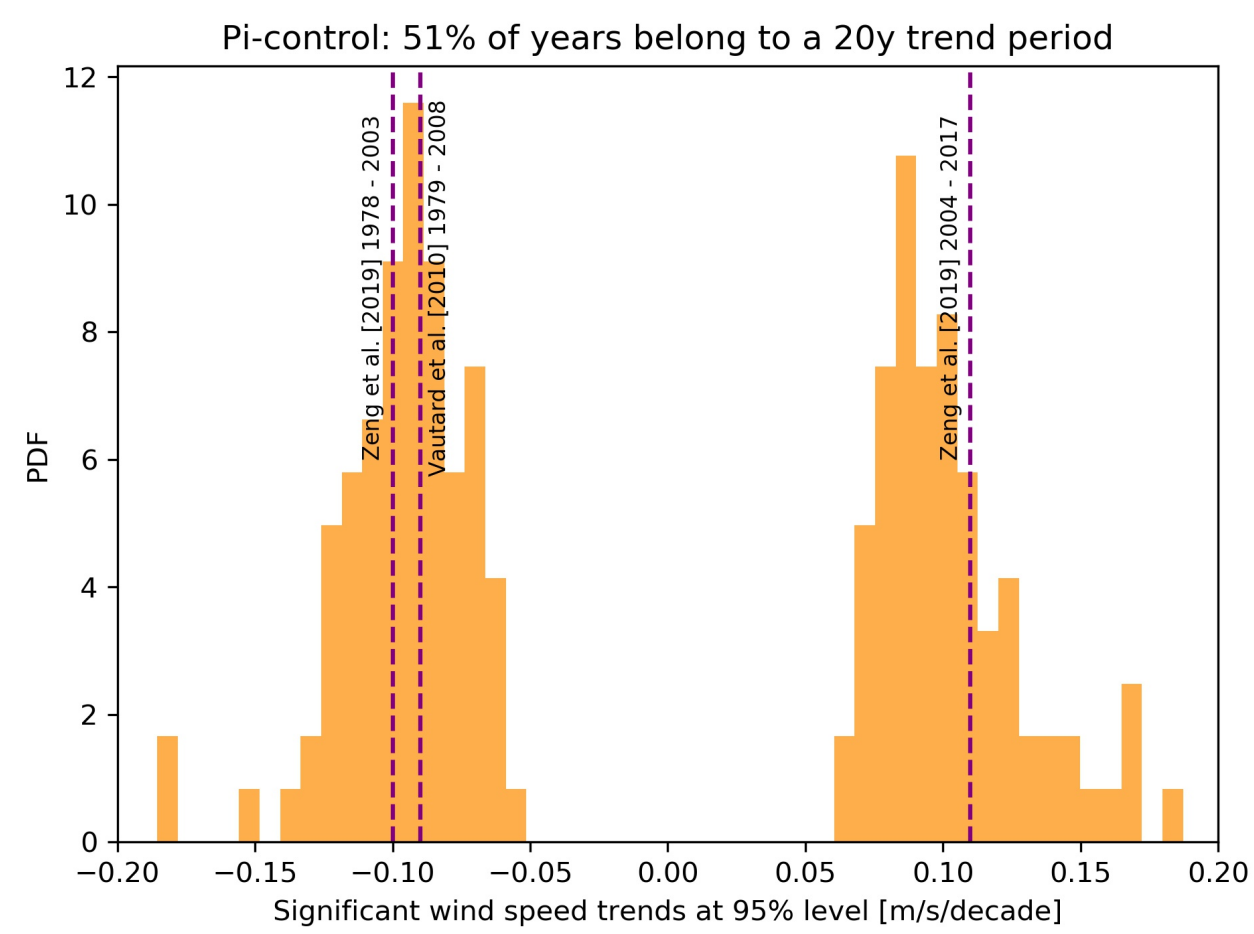

**Figure A2. 20-year trends in annual mean wind speed in MPI-GE averaged to the European station sites in HadISD**. Figure is analogous to 7, except that the average is taken over a box containing Europe but over all HadISD station locations that pass the quality control in (Zeng et al., 2019).

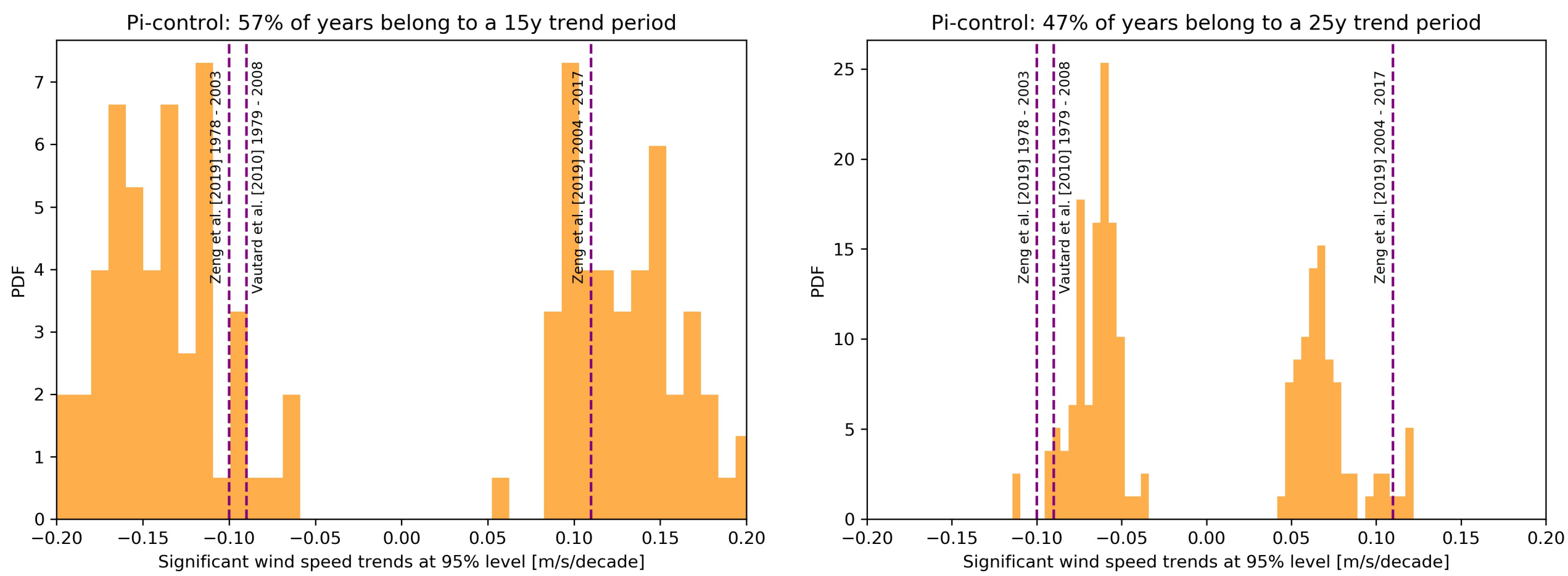

**Figure A3. 15 and 25-year trends in annual mean wind speed in MPI-GE over Europe**. Figure is analogous to 7, except that trends are computed over different time periods.

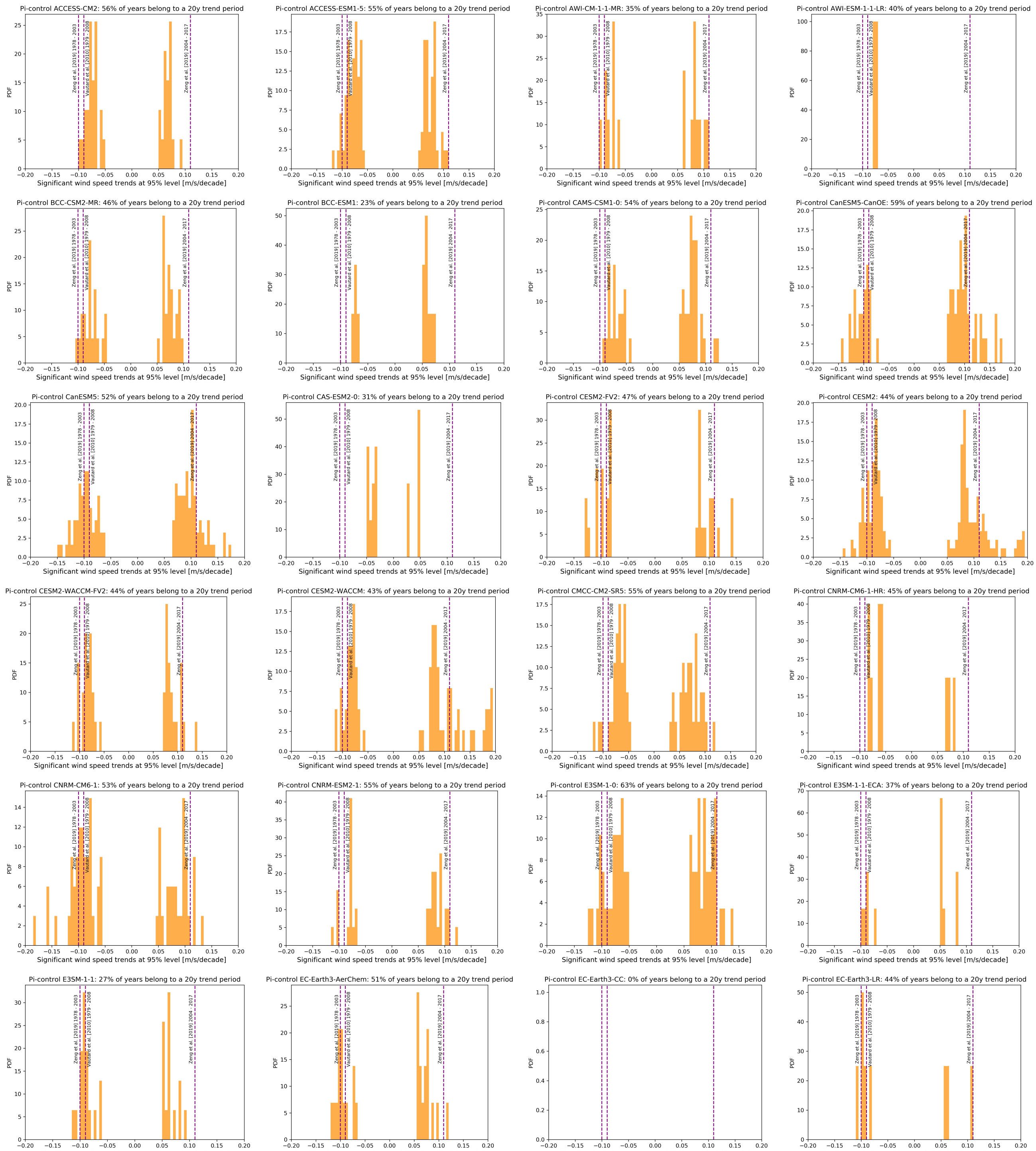

**Figure A4. 20-year trends in European annual mean wind speed in CMIP6.** As in Fig. 7b but for individual models.

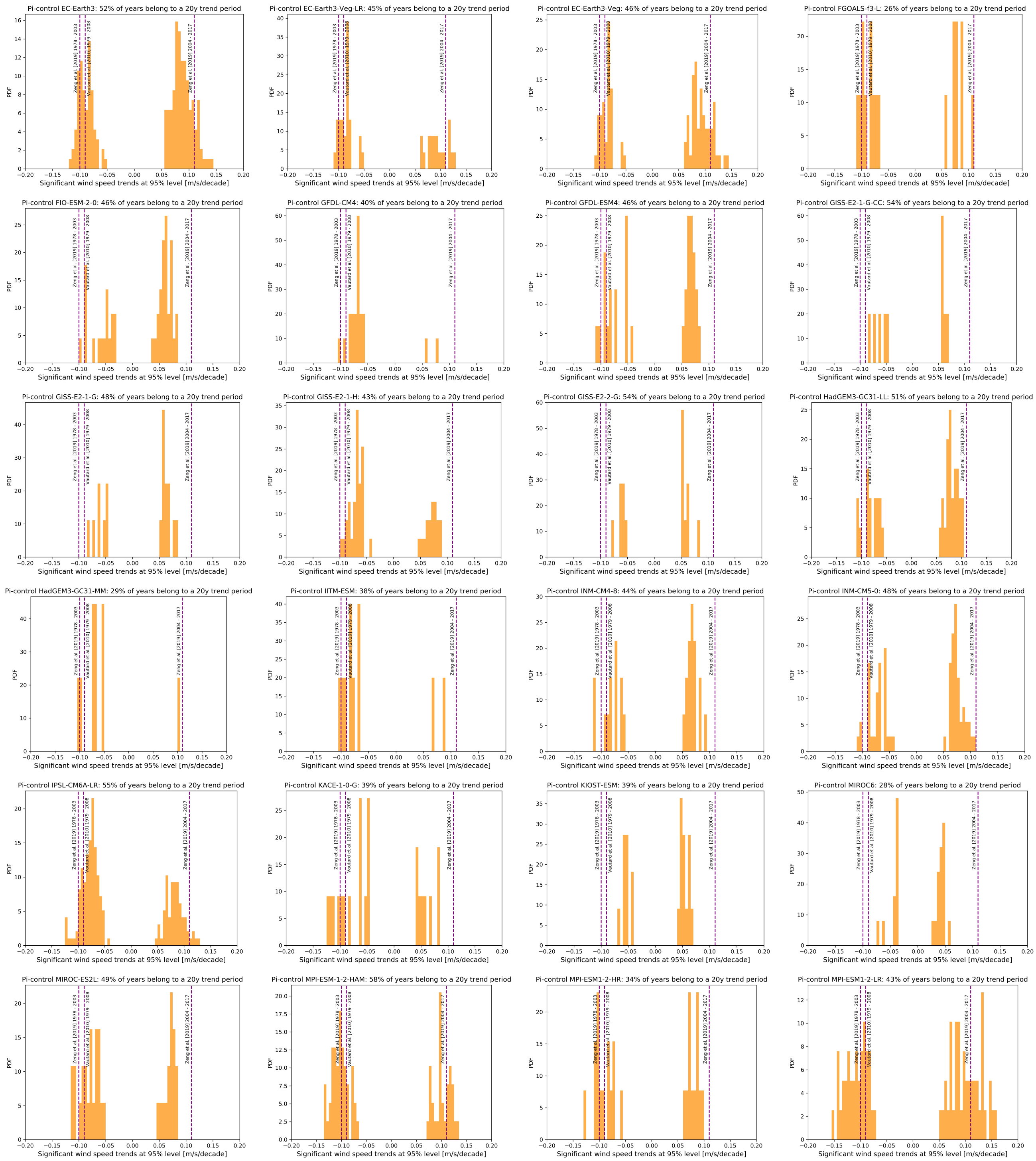

**Figure A5.** Continuation of Fig. A4.

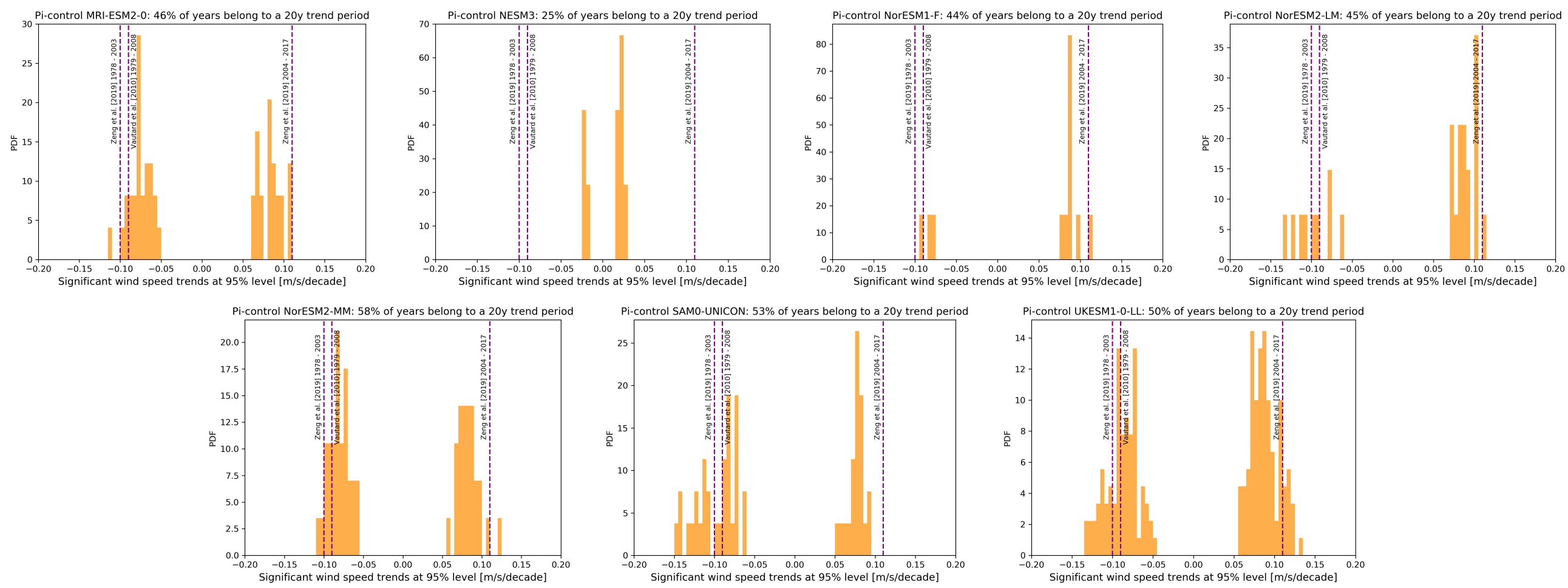

**Figure A6.** Continuation of Fig. A5.