# Peer review of "Wind speed stilling and its recovery due to internal climate variability"

_Earth System Dynamics, 2021_

## Author Comment (AC1)

Wind speed stilling and its recovery due to internal climate variability

Wohland, Jan; Folini, Doris; Pickering, Bryn

Article reference: esd-2021-29

**Response to the reviewers**

We would like to thank Laurent Li and the second anonymous reviewer for their assessment of our manuscript. We are glad both reviewers consider our manuscript to be well written and suggest to accept it after revisions. We believe that the detailed and precise comments helped us a lot to improve the manuscript. Below, we provide point-by-point responses to all comments, including planned changes to the manuscript.

**In this document, *italics* denote quotations from the reviewer assessments.  and blue denote deletions and additions to the manuscript text. Line numbers refer to the initial submission. Following the ESD recommendations, we do not provide a revised version of the manuscript now but will do so after the editor has made a decision.**

**Reviewer 1: Laurent Li**

*This manuscript uses large-ensemble climate simulations to investigate changes of land surface wind speed. Climate simulations are from MPI Grand Ensemble in the CMIP5 framework: a long preindustrial control simulation, historical simulation, future scenarios RCP2.6, 4.5, 8.5, and an idealized simulation with CO2 increase at 1% per year. The use of large ensemble provides the possibility to separate effects of external forcing from those of internal climate variability. The idealized simulation of 1% CO2 increase is used to estimate the effect of CO2-induced global warming (called dynamical contribution in the manuscript), and to deduce a residual term attributable to other contributors (mainly considered as effect of land-use in the manuscript).*

*There are two main conclusions. The first one is on the response of wind speed to external forcing. It was shown that CO2-related global warming has a very small effect for wind stilling, but the historical land-use and future evolution of vegetation cover are the most important contributor for changes of wind speed. The second conclusion is that the internal climate variability plays a major role in explaining the ups and downs of wind speed at the time scale of decades and these decadal changes of wind speed are much larger than effects of external forcing, whatever its nature (CO2 or land-use). These conclusions are in line of my expectations.*

*The manuscript is generally well written. I recommend its acceptance for publication in Earth System Dynamics. I have a few comments that might be useful for the authors to improve their manuscript with a few minor revisions. Some of them are useful for further discussions, so not mandatory to be incorporated in the manuscript.*

**Author response**

We are pleased to read that you consider our manuscript to be in a good shape and that our results align with your expectations.

**Reviewer comment 1-1**

*For the decomposition of wind speed changes into a dynamical contribution and a residual term. I think the authors are well aware of the imperfections of this decomposition, but I am still a little unsatisfied by the ignorance of effects related to aerosols, land-sea thermal contrast (different rapidity of warming), and other non-CO2 greenhouse gases.*

**Author response**

Thanks for this comment related to our approach. Yes, we are aware that there are some imperfections of the method and we would like to stress that we explicitly state the limitations in the manuscript:

- „While this approach provides a reasonable proxy, it is not exact for a few reasons, including the effect of non-CO2 species such as ozone and the stronger emissions in the stylized experiment that leaves the climate system less time to respond to the forcing." (l. 103-105)
- „While these numbers clearly document a link between residual wind speed changes and changes in primary and secondary land, a one-to-one relationship does not exist. Such a relationship, however, is also not expected for two main reasons. First, the effects of land-use changes are not restricted to the immediate vicinity and wind speeds in one grid box

may well be influenced by land-use changes in adjacent grid boxes. Second, the correspondence between the idealized 1%CO2 experiment and the other experiments is only an imperfect proxy for the dynamical change due to, among others, a significantly higher rate of emissions that leaves the climate system less time to respond to the forcing. Another important difference are non-CO2 emissions like ozone and aerosols that are ignored in the idealized 1%CO2 experiment. Nevertheless, we report a very good link between positive residual wind speed changes and land-use change overall." (l. 155 - 162)

Of course, we agree that surface wind speeds can also be altered through the additional processes that you mention. CO2 and land-use change, however, are the most important drivers because we are able to explain most of the wind speed changes by evaluating the two. Nevertheless, we agree that including additional processes would make the analysis more complete and may explain discrepancies in some areas.

For instance, Bichet et al. (2012) found that aerosol emissions can decrease wind speeds in India by up to -0.2 m/s over the period 1870-2005 and this reduction could explain the discrepancy between land-use change and residual change in that area (compare Fig. 1c and 1d).

Given the spatial distribution of the wind speed changes (see Fig. 1a,e,i,m), we do not think that intensified land-sea breezes play an important role: There does not seem to be a strong link between closeness to shore and amplitude of change. Nevertheless, changes in the monsoon circulations in Africa and Asia might contribute to the overall changes in these regions.

Non-CO2 greenhouse gas emissions are a complex topic on their own. Their short lifetime requires to treat them differently than CO2 as the rate of change of emissions rather than cumulative emissions control the forcing. Again, we think that addressing non-CO2 emissions could be an interesting step for future analysis, yet is not compulsory here due to the good match between forcings from land use and CO2, and wind speed.

In total, we believe that it is justified to ignore aerosols, land-sea thermal contrasts and non-CO2 GHGs when answering the research question of the paper under review (Understanding „whether stilling and its reversal are manifestations of internal climate variability or have been forced"). There are also pragmatic reasons related to data availability that hinder expansion of our approach as no equivalent to the 1%CO2 runs exist for non-CO2 GHGs or aerosols such that isolating their effects is more difficult. Nevertheless, we very much agree that in-depth investigations of the mentioned other processes is worthwhile and should be undertaken in future research. We add more clarification and discussion, as detailed below, to emphasize this point.

**Changes to the manuscript**

l. 160 f:
Another important difference are non-CO2 emissions like ozone and aerosols that are ignored in the idealized 1%CO2 experiment. Including non-CO2 emissions could help to explain localized mismatches between the residual change and land-use change, for example, in India where Bichet et al. (2012) found aerosols to reduce wind speeds by up to 0.2 m/s over the period 1870-2005 (cf. Fig. 1c and 1d) . Nevertheless, we report a very good link between positive residual wind speed changes and land-use change overall.

l. 290f:
In addition to the competing effects of land-use change and greenhouse gas emissions examined here, Bichet et al. (2012) find that aerosol emissions play a significant role in some places. To gain a more complete understanding, future studies might want to isolate the effects of aerosols and non-

CO2 greenhouse gas emissions, either through dedicated modelling or once stylized experiments for these forcing agents become available.

**Reviewer comment 1-2**

> *For the methodology. It is not very clear how to deduce trends for the 20-y periods. This is a quite important issue, since the main conclusion of the manuscript is dependent on this procedure. Please also precise if results are robust or sensitive to any choices of tuneable or predefined parameters in the methodology.*

**Author response**

Thanks for making us aware of this issue. We agree that the method was not well documented in this particular instance and expand the documentation as given below. Please note that we test the sensitivity to different trend durations (see Fig. A3) and discuss this sensitivity in lines 215 to 217 („As expected, trends computed over a shorter period occur more often (57% for 15y trends) and have a larger magnitude while longer trends have weaker magnitudes (see Fig. A3). While the specifics vary with trend length, significant trends of similar magnitudes to the observed ones emerge independent of trend length.“). We also tested the effect of different significance levels (90%, 85%) and add another Figure to the Appendix.

**Changes to the manuscript**

l. 109 ff:
We compute linear trends over 20y time periods which is a reasonable timescale for stilling and its reversal given that stilling in the observational datasets spans 25 to 30 years while its reversal currently lasts less than 15 years (Zeng et al., 2019). Trends are referred to as stastically significant if they are different from zero at a 95% significance level based on a Wald test with t-distribution of the test statistic as implemented in the Python module *scipy.stats.lingress,* identical to the approach taken in Wohland et al. (2020). We ensure robustness of the approach by sensitivity tests with different trend durations and significance levels.

l. 212 ff:
These results are robust and remain valid under different spatial sampling, different trend lengths, trend significance levels, and using a large CMIP6 ensemble. (...) As expected, trends computed over a shorter period occur more often (57% for 15y trends) and have a larger magnitude while longer trends have weaker magnitudes (see Fig. A3). The same applies to trends computed with lower significance thresholds (see Fig. A4). While the specifics vary with trend length and significance level, significant trends of similar magnitudes to the observed ones emerge independent of these tunable parameterstrend length.

New Fig. A4:

[Figure]

**Figure A4. 20-year trends in annual mean wind speed in MPI-GE over Europe during pi-control**. Figure is analogous to 7 a, except that trends are only shown if they are significant at the 90% (a) or 85% (b) level.

**Reviewer comment 1-3**

> *The manuscript focuses on wind speed changes in Europe. How about other regions of the world? Asia? North America? Are there any coherent structures across the globe? It seems that observation (e.g. Zeng et al., 2019) also reveals important changes of wind speed in other parts of the world.*

**Author response**

Thanks for this comment. Please note that the first half of the manuscript, where we evaluate the long-term trends due to changes in land use, has global coverage. The results shown in Figures 1 to 5 are based on global data.

We decided to restrict the second half of the paper to the European domain in the interest of conciseness and because we are most familiar with the European context and have assessed multidecadal variability using different datasets in this domain. We do not think that there is any good reason for Europe to be unique in featuring stilling as a consequence of internal climate variability. Instead, it seems likely that multidecadal near-surface wind speed changes in other parts of the world, in particular in the mid-latitudes, are also manifestations of internal climate variability. We add some explanation along these lines to the discussion and encourage in-depth investigations in future research.

**Changes to the manuscript**

l. 258ff:

„To the best of our knowledge, this study is the first to use a large climate model ensemble to understand the wind speed effects of land-use change and wind speed stilling. While we focused on Europe in the second half of the study, we believe that our results are applicable in other parts of the globe, including North America and Asia, where other low-frequency modes of climate variability can generate similar multidecadal fluctuations in surface winds. Our results complement, extend and partly contradict ...“

**Reviewer comment 1-4**

> *Can the authors comment on the origin of internal climate variability? At decadal scale, the global ocean should be the main player. Are there any corresponding variations at global sea surface which can generate such internal variability?*

**Author response**

Thanks for this comment. Yes, we agree that the ocean most likely provides the memory needed for these multidecadal wind fluctuations and it is well established that the North Atlantic undergoes multidecadal changes (e.g., Keenlyside et al., 2015). In fact, we found in an earlier study that multidecadal fluctuations in German wind energy generation have high correlations with the low-frequency component of the North Atlantic Oscillation (Wohland et al., 2019). This link makes much sense since the NAO is a measure of the pressure gradient strength across the eastern North Atlantic and this pressure gradient has large control over wind speeds in Europe. However, the link between the NAO and surface wind speeds varies by location, generally increasing in strength closer to the North Atlantic coast. This spatial pattern gives rise to balancing potential, in particular when combining wind resources in Greece, Portugal and central Europe, as found by Neubacher et al. (2021).

Please note that this aspect is already mentioned in ll. 299ff („This finding is consistent with an emerging body of literature that links multidecadal changes in wind and wind power during the historical period to patterns of mulitdecadal climate variability such as the North Atlantic Oscillation (Wohland et al., 2020; Zeng et al., 2019).“). We thus decided not to discuss this aspect again.

REFERENCES

Keenlyside, N.S., Ba, J., Mecking, J., Omrani, N.-E., Latif, M., Zhang, R., Msadek, R., 2015. North Atlantic Multi-Decadal Variability — Mechanisms and Predictability, in: World Scientific Series on Asia-Pacific Weather and Climate. WORLD SCIENTIFIC, pp. 141–157. https://doi.org/10.1142/9789814579933_0009

Wohland, J., Omrani, N.E., Keenlyside, N., Witthaut, D., 2019. Significant multidecadal variability in German wind energy generation. Wind Energ. Sci. 4, 515–526. https://doi.org/10.5194/wes-4-515-2019

Neubacher, C., Witthaut, D., Wohland, J., 2021. Multi-decadal offshore wind power variability can be mitigated through optimized European allocation. Adv. Geosci. 54, 205–215. https://doi.org/10.5194/adgeo-54-205-2021

**Reviewer comment 1-5**

> *Can the authors comment on the possible mechanism that land-use plays a major role of external driver for changes of surface wind?*

**Author response**
Most importantly, land-use change alters surface roughness and thus surface drag, which affect surface wind speeds locally. There are many other possible mechanism in which the two are connected. For instance, a change in vegetation alters the albedo, temperature and potential for moisture recycling, thereby changing latent and sensible heat fluxes which in turn alter stability and can impact wind speeds on local and regional scales.

Process-level assessments of the interplay of land-use change and near-surface wind speeds would thus be an interesting next step of this analysis. We believe that the CMIP6 data is better suited for such analyses due to the more detailed representation of land-use change in LUH2 compared to LUH1. Again, we would like to stress that the details of these interactions are interesting, yet they are not essential to answer the research question addressed in this study.

**Changes to the manuscript**

l. 156 ff:
„First, the effects of land-use changes are not restricted to the immediate vicinity and wind speeds in one grid box may well be influenced by land-use changes in adjacent grid boxes. In addition to directly impacting surface winds via altered surface roughness, land-use change can also indirectly affect wind speeds, for instance via modifications of latent and sensible heat fluxes caused by changes in albedo, temperature and moisture recycling. Second, ...“

**Reviewer 2:**

*This paper investigates the impact of internal climate variability on trends in near-surface wind speeds in historic and future climates. Therefore the authors investigate wind speed variability in the MPI-GE climate model for historical and future (RCP 2.6, 4.5, and 8.5) scenarios including land-use changes and a reference simulation with increase of CO2 only. The latter allows to assess the contribution of the "dynamical response" to greenhouse gas forcing to the total change in wind speed. The authors find that in the ensemble mean, the residual change highly correlates with land use change, suggesting a strong contribution of land-use change to overall changes in onshore wind speed. However, investigating decadal trends in 20 year sub-periods the authors find that the decadal trends due to internal climate variability are 10-fold higher than the change due to climate change. This finding is corroborated with the CMIP6 ensemble.*

*Recommendation:*

*The paper is overall well written, easy to follow, and the Figures and Table adequate. The study provides novel and insight in the ongoing discussion about "global stilling", clarifying the magnitude of wind speed trends due to internal variability one also needs to expect under climate change. It fits well in the scope of ESD and should be of interest to its readership. I have only one two comment and some small suggestions and recommend to accept after minor revisions.*

**Author response**

We would like to thank reviewer 2 for their thorough assessment. We are glad to read that manuscript and Figures are considered to be good and easy to follow. Please see our answers below for replies to the more detailed comments.

**Reviewer comment 2-1**

*the terminology forced change is a bit confusing and needs to be defined properly. Do you mean due to greenhouse gases, land-use change, wind industry infrastructure or all three? The greenhouse gas effect is likely meant by the dynamical response term only. Thus forced relates to the residual, which should be clarified.*

And

*meaning of forced change with relation to wind related infrastructure is explained too late. Later in the paper you also mean land use change (mainly vegetation?). The use of the term forced change must be clarified.*

**Author response**

Thank you for making us aware of this shortcoming. Please note that we merged two of your comments in the quote above (the first starts in „the terminology" and the second starts with „meaning of") as they deal with the same aspect of the manuscript.

We use the term „forced change" to refer to changes that result from man-made forcing, for instance changes in land use or in atmospheric concentrations. Forced changes are conceptually the opposite of internal variability. While forced changes only occur if there is a trigger (such as cutting down forested areas), internal variability occurs without forcing due to the dynamics of the interconnected climate system. We do agree that the term forced change was insufficiently clarified in the initial version of the manuscript and add additional explanation in the Methods section to avoid similar confusion among future readers.

**Changes to the manuscript**

l. 81 ff:

„In contrast, forcings such as changing greenhouse gas concentrations and land-use change exist in the historical and rcp experiments, and time series represent a superposition of internal variability and a response to the forcings. Throughout this study, we refer to the response to forcings as *forced change*. By contrast, changes that occur as a consequence of internal climate dynamics are referred to as *internal variability*. In the...

**Reviewer comment 2-2**

> *Section 3.2.1 here you relate the internal decadal variability to the "forced changes". While the rate of change for internal variability is nicely shown with data in Fig. 6,7,8 and Europe, the 10x smaller rate for forced changes is not corroborated by data and must be assessed from the global trend in Fig. 5. This is not ideal and a histogram as in Fig. 8 but for the rate of forced changes based on a time-series as in Fig. 5 (but for Europe) would help. (Although this will be a narrow distribution).*

> And

> *Section 3.2.1 Corroborate forced change in future climate with data. See above main comment above.*

**Author response**
Thank you for these two comments that we jointly address here. We believe that that there is a misunderstanding here. The histograms that you are asking for already exists: the green histograms in Fig. 8 (copied below for ease of reading) show the forced changes. From comparison of the orange and green histograms, it becomes apparent that the magnitudes are different by a factor of 10. To avoid similar misunderstandings by others, we have modified the Figure caption to be more explicit and we reference it more clearly in the main text.

[Figure]

Fig. 8a from the manuscript. Green refers to the forced change, orange refers to internal variability. Please see below for the updated caption of Fig. 8.

**Changes to the manuscript**

l. 230 ff:

„Evaluating the MPI-GE ensemble, however, we find that 20y trends of the forced components are too weak to explain stilling roughly by a factor of ten (see Fig. 8). For example, in the historical period, the trends in the forced wind speed changes is at the order of 0.01 m/s/decade (green histogram in Fig. 8a) while the observed trends are one order of magnitude larger (orange histogram in Fig. 8a)."

Caption of Fig. 8:

„20-year trends in European annual mean wind speed in MPI-GE under historic and future climate conditions. Trends are computed for each ensemble member after subtraction of ensemble mean (yellow – representing internal variability) and for the ensemble mean (green – representing forced changes). Different subplots show different experiments. Trends are only shown if they are different from zero at a 95% significance level."

**Reviewer comment 2-3**

> *general comment. It would help to have subsections, describing the data and climate simulations first, then the methodology to quantify different contributions to wind speed change, then the trends.*

**Response to the reviewer**

Thanks for this suggestion. We agree that additional subsections improve legibility and have added them.

**Changes to the manuscript**

l. 64

**„2.1 Climate data sets**

We mainly base…"

l. 80

**„2.2 Seperating forced changes and internal variability**

Different scenarios …"

l 108:

**„2.3 Trends**

We compute..."

*Reviewer comment 2-4*

> *eq (1). The hyphen is a bit confusing, why is it needed?*

**Response to the reviewer**

We chose the hyphen to differentiate between the wind speeds from every ensemble member and the difference in wind speeds to the ensemble mean. One could of couse use a different sign for that (say a hat or a different variable name) but we don't see a reason why a hyphen would be worse than anything else. Note that if you simply remove the hyphen, the equation doesn't make sense any more as $s_i$ on the right hand side and the left hand side would cancel.

**Reviewer comment 2-5**

> *line 106 LUH1 is not defined only LUH in line 70, use consistent abbreviation.*

**Response to the reviewer**

Thanks for spotting this issue, which we fixed.

**Changes to the manuscript**

l. 70

land-use data from Land Use Harmonization (LUH1, Hurtt et al., 2011)

Caption of Fig. 4

Maps show difference between 2090-2100 and 1990-2000 according to LUH1

**Reviewer comment 2-6**

> *Figure 1 and l 136. What is the unit of land-use change? Is it the change in wind speed due to land-use change? If the latter, it must be stated in the Methods how it is calculated?*

**Response to the reviewers**

Thanks for making us aware of this shortcoming. We have added an explanation to the Methods section.

**Changes to the manuscript**

l. 105 ff

„We compare the residual changes $\Delta_{res}$ $s$ to the changes in primary and secondary land in the LUH1 dataset (given as a fraction of the area covered with primary or secondary land) to quantify the effect of land-use change.

**Reviewer comment 2-7**

*Figure 4: remove the subcaption in the subfigures*

**Author response**

Thanks for your suggestion. We believe that you suggest we should remove the titles of the subplots (e.g., IMAGE_rcp26). In our opinion, these titles help the reader understand what is displayed in which subfigure and we thus prefer not to remove them. As far as we know, keeping them does not violate any journal guidelines. Please let us know in case we misunderstood this comment.

**Reviewer comment 2-8**

*Figure 6: it is confusing that the begin/end of an upward/downward trend or of the same direction can happen in consecutive years (or only with a few years apart. This needs a bit more explanation. E.g. must there be pairs of onset upward, onset downward? I.e. is the onset of a downward trend the end of an upward trend?*

**Author responses**

Thanks for these questions. Basically, a red dot denotes the start year of a statistically significant 20y upward trend period and a green dot denotes the start year of a 20y downward period. As we mention in the Methods (lines 114 – 116), this definition implies that consecutive years with red or green dots can occur. To a limited extent, such clustering is visible in Fig. 6, for example around the year 2450 of the pi control simulation.

Regarding your questions:

No, there don't have to be pairs of onset upward and onset downward because the increase might happen as a statistically significant trend over 20y while the decrease is not statistically significant. Similarly, the onset of a downward trend is not needed to mark the end of an upward trend.

We have added a clarifying sentence to the caption.

**Changes to the manuscript**

Caption of Fig. 6

„Wind speed timeseries during pi-control averaged over Europe. Blue (black) lines denotes annual (20y) means and red (green) dots mark onset years of statistically significant upward (downward) trends over 20y periods. Dots of the same color can occur in consecutive years if the trends persist over more than 20y."

**Reviewer comment 2-9**

*Section 3.2 and Figures 6,7 discuss pre-industrial control, correct? This must be stated more explicitly in Text and Caption.*

**Author response**

We agree that Fig. 7 should mention pi-control in its legend (currently this information is only given in the subfigure titles) and we add the information.

Please note that the x-axis of Fig. 6 reads „Year of pi-control simulation" and the caption states „Wind speed timeseries during pi-control averaged over Europe". We decided to make the caption even more explicit, see below.

The text already mentions at some points that 3.2 uses pi-control (e.g., in line 205). Nevertheless, we decide to mention this information early in the paragraph to avoid any confusion.

**Changes to the manuscript**

Caption of Fig. 6

„Wind speed timeseries averaged over Europe under pre-industrial control conditions (pi-control).  "

Caption of Fig. 7

„20-year trends in European annual mean wind speed in MPI-GE (left) and the CMIP6 multi-model mean (right) under pre-industrial control conditions. Trends are..."

l. 200

„ We will now add an analysis of internal climate variability by evaluating fluctuations around the ensemble mean in the pre-industrial control simulation."

---

## Author Response (AR1)

**Reply to the editor**

Dear Somnath,

thanks for your evaluation and comments. We are glad to hear that you are satisfied with our responses to the reviewer's comments.

In this document, your comments are in *italics* and additions to the manuscript are highlighted in blue.

**Major comment:**

*Line 101: If I understand you correctly, the total change is decomposed into dynamical and residual components where the total change is calculated from the RCP runs and the dynamical component is from the 1% CO2 run. Is my understanding correct? If so, how will the decomposition work because the CO2 forcing in the RCP scenarios are all different? I assume there is a simple explanation for this because otherwise the whole analysis will fall apart because the residuals will contain some CO2 forcing signal, less in RCP2.6 and more in RCP8.5.*

**Author response**
Please note that we account for the different forcing in the different RCPs by comparing different time windows of the 1%CO2 simulation with the end decades of the RCP scenarios. Specifically, we calculate the year in which CO2 concentrations in the 1%CO2 run are equal to the final CO2 concentration in the RCPs (see Table 1). For example, the RCP85 CO2 concentrations at the end 21st century are about 935 ppm, which corresponds to the year 1971 in the 1%CO2 simulation. By contrast, RCP26 end of century concentrations are only 420ppm, corresponding to the 1891 concentrations in the 1%CO2 simulation.

**Minor comment 1:**
*Line 106: What do you mean by primary and secondary land and "changes in primary and secondary land"? This is not very clear to me and that is why I am not able to fully comprehend what is shown in Fig 2 column 4. Does primary and secondary land refer to primary and secondary vegetation, respectively, as in line 97?*

**Author Response**
Thanks for bringing up this language consistency issue. We adopted the language from the LUH dataset description where „*Secondary* refers to land previously disturbed by human activities and recovering, while *primary* refers to land previously undisturbed by human activities in GLM, both since the beginning of the historical simulation." [Hurtt et al., 2011]. In other words, primary land is the fraction of the grid cell that is covered by undisturbed vegetation (i.e., primary vegetation) while secondary land is the fraction of the grid cell that is covered by vegetation recovering from disturbances (i.e., secondary vegetation).

We decide to link the lines 97 and 106 more clearly to avoid similar confusion in the future.

**Changes to the manuscript**
l. 97:

in all locations that contain either primary or secondary vegetation. While primary vegetation refers to vegetation that has been previously undisturbed, secondary vegetation refers to vegetation that recovers from human intervention. Following the LUH naming convention from Hurtt et al. (2011), we use the term primary (secondary) land to refer to the fraction of a grid cell that is covered by primary (secondary) vegetation.

**Minor comment 2**
Line 168: Please explain in the text what the IMAGE, MiniCAM and MESSAGE acronyms stand for.

**Changes to the manuscript**

Caption of Fig. 4

> **in the rcp scenarios.** The subplot titles give the acronym of the Integrated Assessment Model followed by the name of the rcp. IMAGE stands for Integrated Model to Assess the Global Environment; MiniCAM is the Mini-Climate Assessment Model; MESSAGE is the Model for Energy Supply Strategy Alternatives and their General Environmental Impact. Maps show ...

**Additional author comment**

In a final internal review, we noticed that in the RCP2.6 ensemble members 28-33 and 36-40 are identical. We re-downloaded the data to ensure that the problem wasn't created by our preprocessesing and we informed the dataset providers at MPI about the issue (no answer since early August).

We checked whether the duplicates affect our results, and their impact is too small to matter. For instance, we repeated the trend calculation with the subset of ensemble members that are distinct (i.e., from the total of 100 ensemble members, we exluded the 10 members that are dubplicates). As displayed in the Figures below, both the probability of trend occurrence (51% vs. 53%) and the distribution of the trends relative to the reported number from the literature are very similar. Since the issue is also limited to RCP26 and does not affect the pre-industrial control, historical, rcp45 and rcp85 simulations, we consider it justified to ignore the duplicates here. However, we added a sentence to the Methods section to clearly and transparently flag the issue.

[Figure]

**Full ensemble (N=100)**            **reduced ensemble (N=90)**

**Changes to the manuscript**
l. 71 ff:
> Out of the total 500 ensemble members (5 experiments times 100 members), three members were excluded from the analysis as a cautionary measure because they had a dozen

duplicate time steps. Moreover, we found that ensemble members 28-33 and 36-40 have identical wind speeds in rcp2.6. These duplicates only have negligible effects on our results which we verified by repeating the analysis with a subset of members that are mutually distinct. For internal consistency, and since the other scenarios are not affected, we decided to always use the full ensemble.

Kind regards,
Jan (for all authors)